



# Potential of X-band polarimetric SAR co-polar phase difference for Arctic snow depth estimation

Joëlle Voglimacci-Stephanopoli[1,2], Anna Wendleder[3], Hugues Lantuit[4,5], Alexandre Langlois[1,2], Samuel Stettner[3,6], Jean-Pierre Dedieu[7,2], Achim Roth[3] and Alain Royer[1,2]

[1] Centre d'Applications et de Recherches en Télédétection, Université de Sherbrooke, Sherbrooke, J1K 2R1, Canada
[2] Centre d'Études Nordiques, Université Laval, Québec, Québec, G1V 0A6, Canada
[3] German Remote Sensing Data Center, German Aerospace Center, Oberpfaffenhofen, Germany
[4] Institute of Geosciences, University of Potsdam, Potsdam, Germany
[5] Alfred Wegener Institute Helmholtz Centre for Polar and Marine Research, 14473 Potsdam, Germany
[6] German Space Agency, German Aerospace Center, Bonn, Germany
[7] Institute of Environmental Geosciences, Université Grenoble-Alpes/CNRS/IRD, 38058 Grenoble, France

*Corresponding author:* Joëlle Voglimacci-Stephanopoli (joelle.voglimacci-stephanopoli@usherbrooke.ca)

**Abstract.** Changes in snowpack associated with climatic warming has drastic impacts on surface energy balance in the cryosphere. Yet, traditional monitoring techniques, such as punctual measurements in the field, do not cover the full snowpack spatial and temporal variability, which hampers efforts to upscale measurements to the global scale. This variability is one of the primary constraints in model development. In terms of spatial resolution, active microwaves (synthetic aperture radar—SAR) can address the issue and outperform methods based on passive microwaves. Thus, high spatial resolution monitoring of snow depth (SD) would allow for better parameterization of local processes that drive the spatial variability of snow. The overall objective of this study is to evaluate the potential of the TerraSAR-X (TSX) SAR sensor and the wave co-polar phase difference (CPD) method for characterizing snow cover at high spatial resolution. Consequently, we first (1) quantified the spatio-temporal variability of the geophysical properties of the snowpack in an Arctic catchment, we then (2) studied the links between snow properties and CPD, considering ground vegetation. Snow depth (SD) could be extracted using the CPD when certain conditions are met. A high incidence angle (> 30 °) with a high Topographic Wetness Index (TWI) (> 7.0) showed correlation between SD and CPD (R-squared up to 0.72). Further, future work should address a threshold of sensitivity to TWI and incidence angle to map snow depth in such environments and assess the potential of using interpolation tools to fill in gaps in SD information on drier vegetation types.

## 1. Introduction

Snow cover is a key component of the cryosphere which plays an essential role for ecological processes and hydrological dynamics. In arctic ecosystems, those processes include species survival (Dolant et al., 2018; Poirier et al., 2019), thermal ground regime (Goodrich, 1982; Gouttevin et al., 2012; Stieglitz et al., 2003) or vegetation colonization and growth (Berteaux et al., 2017; Kankaanpää et al., 2018; Myers-Smith et al., 2011a). In the past 40 years, we observed a pan-Arctic reduction in



the snow cover duration of 2–4 days per decade (AMAP, 2017) and maximum Arctic snow depth trend show a consistent decrease since 1980 (AMAP, 2017; IPCC, 2019). These trends will undeniably change the arctic landscape. For instance, duration of snow patches impacts vegetation phenology (Kankaanpää et al., 2018) and controls shrubs' growth (Myers-Smith

et al., 2011b; Pomeroy et al., 2006). Hence, patterns of vegetation densification (also called greening) arise, and dense vegetation such as shrubs impact the snowpack physical properties. Twigs induce a decrease in snow density and an increase of depth hoar formation (Domine et al., 2016; Gouttevin et al., 2018; Sturm et al., 2001). By protruding above the snowpack surface, shrubs reduce surface albedo and advance the snow melt timing (Sturm et al., 2001). Coupled to a decreasing trend on maximum snow depth and snow cover duration observed (AMAP, 2017; IPCC, 2019), the greening of the Arctic is likely

to lead to drastic modification of the snowpack. A recent update on the classification of Sturm et al. (1995) suggested by Royer et al. (2021) demonstrates a positive feedback on climate warming owed to snow-vegetation interaction. High resolution land cover classification is therefore needed to address changes in the snowpack in a warming climate.

Current snow modules used in Earth System Models are based on coarse spatial resolution of tens of kilometres (Bokhorst et al., 2016). Coarse special resolution hampers our efforts to understand the dynamics driving snowpacks at the landscape scale.

Indeed, snow is characterized by a high spatial and temporal heterogeneity ( *e.g.*: Rutter et al., 2014; Thompson et al., 2016; Wilcox et al., 2019). Traditional approaches using *in situ* measurement can provide very detailed spatial information on snow properties, but cannot be deployed over large areas. There is therefore a strong need to bridge these two scales and provide means to monitor the temporal and spatial variability of the snowpack over larger areas.

Earth observation satellites can provide frequent measurements over larger areas. Space borne platforms are widely used to

monitor snow on local, regional, and global scales. Yet, they also suffer from strong limitations. Optical sensors allow measurements on surface characteristics of snow, but do not provide direct measurements of the properties of the snowpack and are often limited by cloud cover. Passive microwave monitoring methods are operational and provide continuous data, but suffer from the coarse spatial resolution of satellite observations (*e.g.*: Frei et al., 2012)). Active microwave observations with synthetic aperture radar (SAR) can overcome these issues in providing high resolution frequent snow measurements over large

areas.

SAR can "see" through clouds while being independent from solar illumination. SAR sensors are interesting to collect data from the snowpack because they can, on the one hand, transmit and receive microwaves in horizontal (H) and vertical (V) polarization, and on the other hand, their microwaves can interact with and penetrate into the observed material. The main challenge related to the use of SAR is the lack of a reliable method to relate satellite data to physical measurements in snow-

impacted environments.

The objective of this paper is therefore to evaluate the potential of polarimetric method co-polar phase difference (CPD) produced with the X-band satellite TerraSAR-X to retrieve SD from an arctic snowpack where vegetation is highly variable. This general objective requires a complete characterization of the snowpack from field data to fully understand the sensitivity of CPD to various snow characteristics. This requirement motivates the following two specific objectives: (1) investigate SD

variability between different vegetation classes in the Ice Creek catchment (Qikiqtaruk-Herschel Island, Yukon, Canada) using



*in situ* measurements collected over the course of a field campaign in 2019 and (2) evaluate linkages between SD and CPD distributions considering meteorological data over the 2015–2019 period.

## 2. Background: Co-polar phase difference—snow structure

### 2.1. Arctic snow properties

Snow cover in the Arctic is mostly characterized by two main layers (Domine et al., 2016; Royer et al., 2021; Sturm et al., 2008). The upper layer, the wind slab, is very compact as it is subject to sustained winds and cold temperatures that promote cohesion of snow grains (Domine et al., 2018b; Sturm et al., 2008). The basal layer generally consists of depth hoar (DH) grains that develop under a kinetic metamorphic regime in dry snow conditions with a sustained strong temperature gradient (Domine et al., 2016).

The snowpack is driven by two types of metamorphic regimes, namely wet and dry snow metamorphism (Bernier et al., 2016). These regimes develop according to the temperature gradient in the snowpack and to its liquid water content (Colbeck, 1973). Wet snow metamorphism, with liquid water available in the snowpack, will lead to different metamorphic processes for saturated and unsaturated conditions (Colbeck, 1982). As a result, there will be a major impact on microwave radiative transfer given that wet snow acts as a blackbody in such frequencies (*e.g.* Rott and Matzler, 1987). In the case of an arctic snowpack,

a regime of dry snow metamorphism is generally found when sustained cold temperatures last during most of the winter (Domine et al., 2018a). Schneebeli et Sokratov (2004) found that snow crystals are highly anisotropic (dependency to a direction of an object), which is correlated with snow metamorphism (Calonne and al., 2014; Gouttevin and al., 2018). As such, over the time, snow crystals become elongated to a vertical direction after their setting up in the snowpack.

The geometrical structure of the snow will characterize the electromagnetic wave propagation through the snowpack by

scattering and absorption processes within each layer (Mätzler, 1987). Given the dry nature of the arctic snowpack, the main source of backscattering should occur at the snow-ground interface for frequencies in X-band ($\lambda = 3.1$ cm) such as used in this study or below as dry snow can be considered as a homogeneous, "non-scattering" and non-absorbing volume (Leinss et al., 2014). This said, inhomogeneous layers such as ice layers, melt/freeze crust and any strong vertical change in dielectric properties (*i.e.* density, wetness) can also affect the signal.

### 2.3. Co-polar coherence

The co-polar coherence (CCOH) indicates the correlation coefficient of HH and VV phase centers. The magnitude of the CCOH ranges between 0 and 1 where a weak correlation ($< 0.5$ as defined by Leinss et al. [2014]) indicates a low scattering with a more chaotic and randomly phase shifts between HH and VV waves and are hence omitted. Such weak correlations will occur when HH and VV waves have different phase centers and different scattering targets. A decrease in correlation can also

be induced by a strong surface scattering caused by rough or wet surfaces, volume scattering during winter or during snow-




free conditions where vegetation is exposed (Fig.1). The equation of CPD is only valid where no volume scattering occurs (Leinss et al., 2014) since an increase in volume scattering will lead to a decrease of the CCOH.

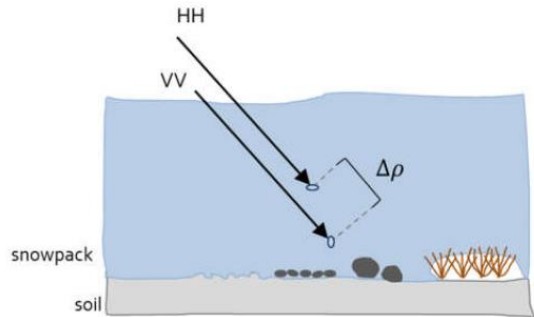

**Figure 1.** Phase shift can also be caused by scattering effects within the snowpack or by surface roughness (including vegetation) (Credit: A.Wendleder).

### 2.4. Co-polar phase difference

CPD is a polarimetric method using difference in the phase between HH and VV polarization channels. The phase difference refers to the difference in the propagation speed of a wavelength in a material as a function of polarization, which then causes a phase difference in the electromagnetic wave between polarizations. The phase of a single polarization is assumed to have a uniform distribution over [-π, π] (Leinss et al., 2014; Patil et al., 2020).

A relationship was found between CPD and snowfall by Chang et al. (1996) and Leinss et al. (2014) which induces a propagation delay among horizontal and vertical phases due to horizontal alignments of fresh snow crystals. Recent studies focused on the boreal region (Leinss et al., 2014, 2016) or were applied in arctic region with no or sparse vegetation (Dedieu et al., 2018) so the application of the CPD method in the Arctic remains poorly documented. It could be hypothesized that the CPD can describe the entire snowpack in such cold and dry environments. Strong vertical changes in density and grain size could also lead to a decrease in coherence so that the use of CPD information might not be suitable (*i.e.* when CCOH < 0.5, see Fig. 1).

## 3. Data and Methods

### 3.1. Study site

Qikiqtaruk-Herschel Island (69° 35' N, 139° 06' W) is located about 2 km off the Yukon Coast in the northwestern Canadian Arctic (Fig. 2). With an approximate area of 108 km$^2$, this island has a rolling topography (max. altitude: 183 m a.s.l.), dissected by numerous geomorphological forms such as gullies, valleys and polygonal soils (Short et al., 2011; Stettner et al., 2018). The permafrost on Qikiqtaruk-Herschel Island is continuous with a high ice content. Ground ice can be observed on the island in the form of ice wedges, ice lenses, or buried snowbanks, as observed by Pollard (1990). Results from Wolter et al. (2016) suggested that geomorphological processes, such as permafrost degradation, are strongly related to vegetation



composition on Qikiqtaruk-Herschel. The active layer thickness varies from 45 cm to 90 cm in marine deposits (silty diamicton) and can reach over a 110 cm in porous deposits (Lantuit and Pollard, 2008; Smith et al., 1989). A thickening of the active layer by 15 cm to 25 cm was documented on the island during the period from 1985 to 2005, as well as an increase in the mean annual air temperature by 2.7 °C between 1970 and 2005 (Burn and Zhang, 2009).

Spatial distribution of snow on the island is primarily based on topography, due to the low tundra-type vegetation (Burn and Zhang, 2009). The snow is blown away from the uplands and accumulates in topographic depressions such as valleys and hummocky terrain (Burn and Zhang, 2009). The dominant wind direction is northwest with frequent storms in late August and September (Solomon, 2005). A study by Myers-Smith et al. (2011) indicates an increase in the canopy and vegetation height over the last century that can be expected to have an impact on the snow cover structure.

In this paper, we performed our measurements in the Ice Creek catchment (area of 1.54 km$^2$) located at the eastern end of the island (Fig. 2). The digital elevation models from ArcticDEM (2 m res.) indicates average slopes of 2.9° with maximums of 13.2° at altitudes ranging from 5 m to 94 m (Porter et al., 2018).

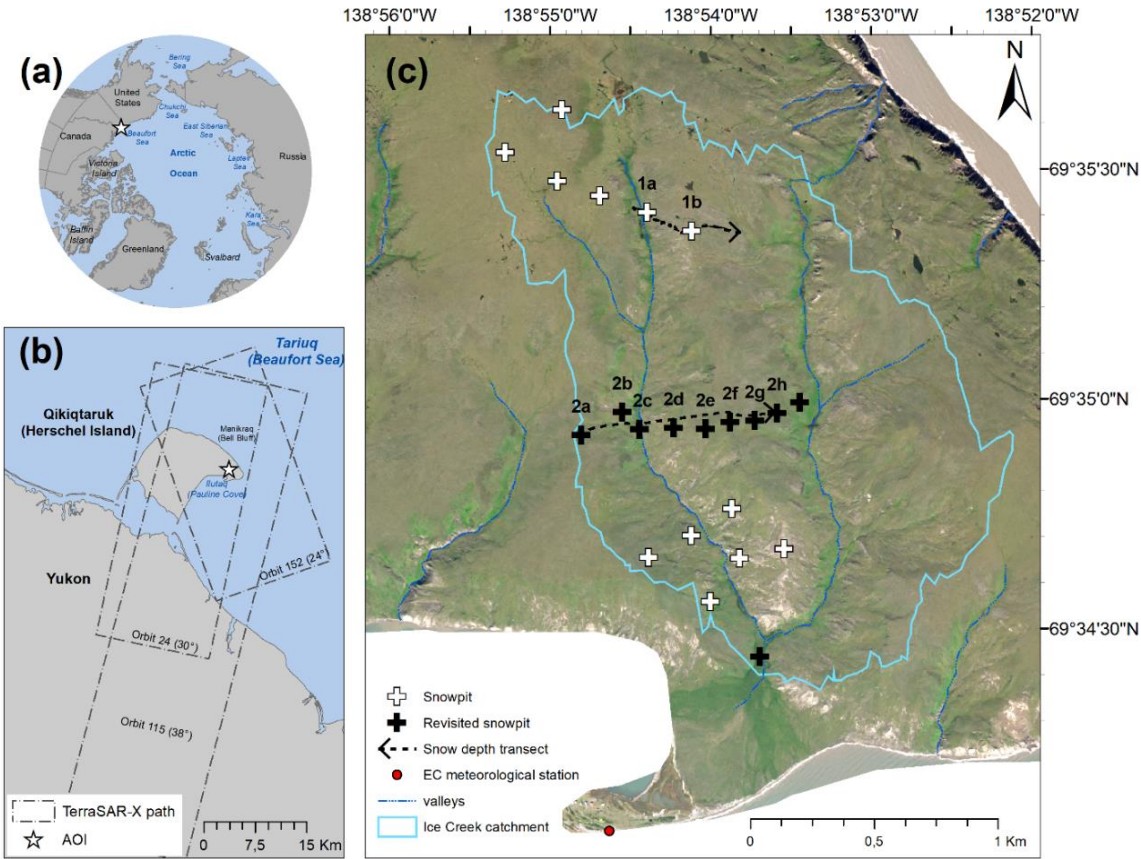

**Figure 2.** (a) Location of the study site in the Arctic (b) Visual extent of TerraSAR-X passages on Qikiqtaruk Herschel Island (c) Ice Creek
study site including the location of measurements. The black crosses were revisited during each TerraSAR-X acquisitions (Imagery provided by Worldview 01.01.1001, True Color). The meteorological station belongs to Environment and Climate Change Canada (ECCC).



### 3.2. Snow distribution over Ice creek

#### 3.2.1. Snow measurements

Two sampling strategies were used for the snowpit characterization (Table 1). First, detailed snowpit measurements were
conducted along predefined locations at an average distance of 200 m between each site (Fig. 2c). The snowpit locations in
the centre of the Ice Creek catchment as well as location at the outlet of the catchment were revisited during each TerraSAR-
X (TSX, see 3.3.) acquisition so that soil characteristics remain unchanged between snow sampling and satellite measurements.
Snow depths were measured using a GPS snow depth probe around the snowpits, ensuring the representativeness of the snowpit
location. This was conducted by measuring depths in a growing circle moving away from the snowpit location until an
approximate diameter of 30 m was reached, which is typically the area required to ensure representativeness in tundra
environments (Clark et al., 2011). Snowpits and SD measurements were then distributed spatially elsewhere in the catchment
to refine the characterization of snow within the catchment. Additionally, two SD transects were conducted across the
catchment to analyze the SD distribution in the study site. Both transects were established from the east side to the west side
of the Ice Creek catchment.
Detailed snow profiles were acquired in spring 2019 (mid-April to early May). In each site, we dug snowpits in a way to avoid
direct solar illumination of the snow wall. High resolution vertical profiles of density, temperature, grain size and type were
conducted according to Fierz et al. (2009, see Table 1). Specifically, layered density profiles were obtained by extracting snow
samples from each identified layer using a 100 cm$^3$ density cutter and weighed using a Pesola light series scale. Temperature
profiles were measured at 3 cm intervals using a Cooper digital thermometer, and profile measurements included shadowed
surface temperature as well as soil-snow interface.
From the above observations, each layer was classified according to their density and snow grain type across 5 classes
following Fierz et al. (2009): 1) Depth Hoar, 2) windslab, 3) surface hoar, 4) fresh snow, 5) melt-freeze crust and ice layer.
The snow depth, mean density of each layer classified, was compiled for later linear regression analysis with TSX data.






**Table 1:** *In situ* **measurements during the 2019 field campaign**

| **Snowpits** | See Fig 2c |
|---|---|
| Stratigraphy | Snow height |
| | Size and grain type (visual estimation) |
| | Temperature profile (measurement at 3 cm, $\pm$ 0.1 °C) |
| | Snow density by layers (measurements at 5 cm when possible, $\pm$ 0.5 kg m$^{-3}$) |
| **Environment and Climate Canada (ECCC) meteorological station** | 69.5682° N, 138.9134° W |
| | Wind speeds at 10 m and 2 m (ms$^{-1}$, hourly) |
| | Precipitation gauge for total precipitation (mm) and rate (mm h$^{-1}$), |
| | Temperature (°C) and relative humidity (%). |
| | Datalogger — Campbell Scientific CR3000E |

### 3.2.2. Vegetation units

The classification of the different vegetation units was obtained from Eischeid (2015) following the initial definition developed by Smith et. al. (1989). The classification was determined by the soil type, vegetation observed and geomorphological features.

The dataset used in this study was derived from 2015 GeoEye satellite data (res.: 1.65 m) (Eischeid, 2015). For the specific needs of this paper, we focused on the following specific classes: *Arctic Willow and Dryas Vetch* (hereinafter referred as *Dryas*), *Arctic Willow and Lupine* (*Lupine*), *Shrub Zone (Shrub)* and *Willow Saxifrage Coltsfoot* (*Coltsfoot*). These classes were selected given that they are physically and spatially different (see Fig. 3d), which is of primary importance from a snow microstructure and radar backscattering perspective.

The *Lupine* class is associated with an irregular and hummocky terrain (Eischeid, 2015, see Fig. 3d). High variability in microtopography results in equally heterogeneous SD at a similar scale (Sturm and Holmgren, 1994). Erosion rates and moisture content will vary greatly following terrain instability (Eischeid, 2015). The *Coltsfoot* class is common in wetlands, where the ground is generally saturated and composed of shrubs (Eischeid, 2015). This vegetation class is located at the bottom of valleys, which is suitable for snow accumulation (Burn and Zhang, 2009). The *Shrub* class was added by Eischeid (2015).

to the original classification by Smith et al. (1989) to reflect the growing importance of shrubs on the island. It is characterized by non-hydrophilic vegetation with lower soil moisture. Finally, the *Dryas* class is common on the gently undulating upland slopes (Smith et al., 1989). The associated soil type is a moderately well-drained Turbic Cryosol, which shows evidence of cryoturbation, as well as bare soil. Each snowpit characteristics and SD measurement were grouped by vegetation units to extract means and standard deviation by vegetation classes. The snowpits made along the two transects were grouped when





they were at a distance less than 30 m and statistics of distribution (average and standard deviation) were extracted to complete

the data analysis.

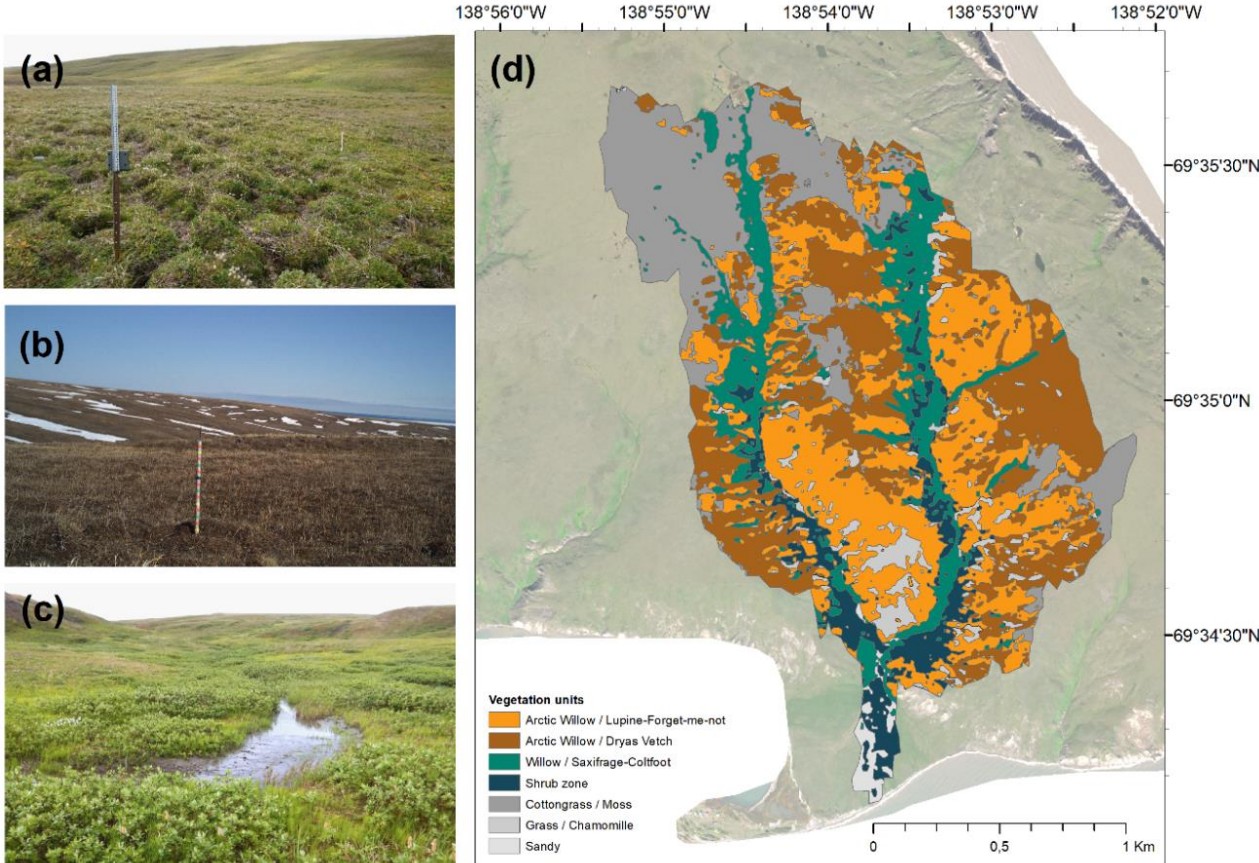

**Figure 3.** Vegetation classes occurring in the Ice Creek catchment: (a) irregular and hummocky terrain observed in *Lupine* class (Credit: M.
Fritz) (b) Low vegetation on well drained area (such as *Dryas*) (Photo by the author) and (c) *Shrub* and wetland (such as *Coltsfoot* class)
(Credit: M. Fritz). (d) Vegetations unit's distribution in the study area (from Obu et al. (2017) as defined initially by Smith et al. (1989). The
classes in grey are not include in the analysis.

### 3.3. Snow- SAR correlation

#### 3.3.1. SAR acquisition and preprocessing

A total of five TSX acquisitions in HH and VV polarizations over three different orbits were obtained during spring 2019,
encompassing areas where snow measurements and vegetation information was available (Table 2). Snowpits and SD
measurements taken before and after (± 2 days) each TSX acquisition were included in the analysis as no precipitation occurred
and air temperature was stable during the field campaign. Additionally, a time series of TSX acquisitions for the 2014–2019
period (orbit 24, θ= 31°) was analyzed to evaluate the inter-annual variability of snow conditions on the island. The full TSX





dataset was first processed at the DLR (German Aerospace Center). The preprocessing is described in Schmitt et al. (2015),
and includes the determination of the Kennaugh elements, their radiometric calibration and orthorectification. The images
were georeferenced in UTM with a ground sampling distance of 5 m. To reduce speckle noise, we used the multi-scale multi-
looking algorithm developed by Schmitt (2016; Schmitt et al., 2015). CPD and CCOH can directly be derived from the
radiometric and geometrically calibrated Kennaugh elements. The Kennaugh matrix describes the polarimetric information
and allows to differentiate the physical scattering mechanisms (*e.g.* double bounce, volume and surface scattering) affecting
the signal, which in turn can be linked to snow characteristics. The following Kennaugh elements were used in the CPD
equation:

$$\phi_{HH} - \phi_{VV} = \text{atan}\left(\frac{K_7}{-K_3}\right) \tag{1}$$

where $K_7$ is the phase shift between HH, and VV phase centre described as

$$K_7 = \text{Im}\{S_{HH}S_{VV}{}^*\} \tag{2}$$

and where $K_3$ is the scattering difference between surface to double bounce:

$$K_3 = -\text{Re}\{S_{HH}S_{VV}{}^*\} \tag{3}$$

To reduce the loss of coherence, a threshold was applied to CPD pixels where CCOH was less than 0.5, following Leinss et
al. (2014). Again, the Kennaugh elements were used in the CCOH equation:

$$\gamma_{VV,HH} \cdot e^{i\phi_{CPD}^\gamma} = \frac{\langle S_{VV} \cdot S_{HH}^* \rangle}{\sqrt{\langle |S_{VV}|^2 \rangle \cdot \langle |S_{HH}|^2 \rangle}} \approx 2 \cdot \sqrt{\frac{K_3{}^2 + K_7{}^2}{K_0{}^2 - K_4{}^2}} \tag{4}$$


A total of 32 pixels had a CCOH less than 0.5, hence showing a random phase shift between waves which is not optimal for
CPD applications. These pixels were therefore removed from the analysis. To discard the potential effect of slope on crystal
grains orientation, 5 pixels with a slope greater than 10° were subsequently extracted (3 in the *Dryas* class, 2 in the *Lupine*).
To assess the temporal variability of CPD signal, pixels were divided by vegetation class for the period 2015–2019.


Table 2: TSX acquisition on Qikiqtaruk-Herschel Island. All orbits were used for linear regression with *in situ* snow measurements.
Orbit 24 has a sufficient time series and was used to extract temporal evolution of CPD.

| Relative orbit | Flight direction | Polarization mode | Incidence angle | Observation period | Number of scenes | *In situ* data |
|---|---|---|---|---|---|---|
| **24** | Descending | HH, VV | 31° | 2014.12.26—2018.03.06<br>2019.04.17—2019.05.20 | 104 | 2019.04.17<br>2019.04.28 |
| **152** | Ascending | HH, VV | 24° | 2019.04.15—2019.05.18 | 24 | 2019.04.26 |
| **115** | Descending | HH, VV | 38° | 2019.04.23—2019.05.15 | 24 | 2019.04.23<br>2019.05.04 |



### 3.3.2. Linking snow depth to CPD

*Implication of Snow Geometry*

We focused on the SD variability between vegetation classes. We also evaluated depth hoar fraction (DHF) given that King et al. (2018) found that X-band backscattering is highly sensitive to depth hoar grains. This allowed us to assess if any discrepancies in SD retrieval can be linked to large grain size. In addition, horizontal structures such as ice layers and melt/freeze crusts were identified for the same purpose of testing the SD retrieval capabilities in different stratigraphic contexts. Dedieu et al. (2018) showed that the attenuation of the SAR signal was caused by ice layers of 3 to 5 cm thick, but lingering

uncertainties remain with regards to the contribution of thinner ice lenses such as the ones found on Qikiqtaruk-Herschel Island.

*Topographic Wetness Index as a proxy*

SD retrieval is challenging because it is impacted by snow surface properties. SD retrieval with CPD may be impacted by the dielectric properties of the snow surface, since the main backscatter signal is expected at the snow ground interface. High

moisture content at the soil surface would potentially improve the performance of SD retrieval, because the presence of ice leads to better reflection conditions for the microwave. The Topographic Wetness Index (TWI) was chosen as a proxy to analyze the variance between vegetation groups. Given the high sensitivity of microwaves to wetness, the high variability of TWI between each vegetation class will lead to different responses in backscattering through changes in the dielectric constant of the soil. The TWI was first developed by Beven and Kirkby (1979) within the runoff model TOPMODEL using the

following equation:

$$TWI = \ln(\frac{a}{\tan\beta}) \tag{5}$$

where $\tan\beta$ is the local slope and $a$ is the upslope area per unit which is obtained with the upslope area (the cells contributing to the runoff to the cells of interest, A) and the contour length (L) following $a=A/L$. The upslope area calculated is based on D8 flow direction algorithm (O'Callaghan and Mark, 1984) and the TWI values were computed based on the ArticDEM

constructed from the DigitalGlobe Constellation (Porter et al., 2018, res: 2 m). Each TWI values derived from the catchment was combined to vegetation classes and CPD cells as described above.

*Analysis*

The Shapiro-Wilk test was used to test the normality of distributions for SD and TWI. Since TWI and SD distributions did not respect a normal distribution, the variance in TWI and SD between each group was tested with the non-parametric test Welch

ANOVA in conjunction with a post-hoc Games-Howell test. We use the Games-Howell test as it does not assume equal variances and sample sizes (Games and Howell, 1976).

We evaluated the correlation between snow characteristics and CPD using a linear regression analysis. The median value was extracted when more than one snow measurement was found in the same TSX pixel (5 m). Thus, a total of 371 pixels was used in the analysis (average number of snow measurements per pixel: 1.7). The median SD by pixels were grouped by vegetation



classes and orbit. Durbin-Watson's test and Breusch-Pagan's were used to assess autocorrelations and homoscedasticity of distribution data. Significance for all tests were calculated with $\alpha = 0.05$.

## 4. Results

### 4.1. Snow distribution

### 4.1.1. Snow depth and depth hoar fraction along transects

Measurements of SD from transect #1 (Fig. 4) varied between 20 and 250 cm where the peak was measured at the valley bottom. Further west, SD values decreased significantly on the slope with values between 20 and 50 cm. Highest DHF along the transect were found on the west side of the transect and on the slopes with an average of 0.76 while an average of 0.39 was observed on the east side of the catchment. Along transect #2 (Fig. 4), and snow cover was also deeper at the bottom of the valley (from 120 cm up to 200 cm) and decreased significantly on slopes and higher elevation areas (30 cm to 50 cm).

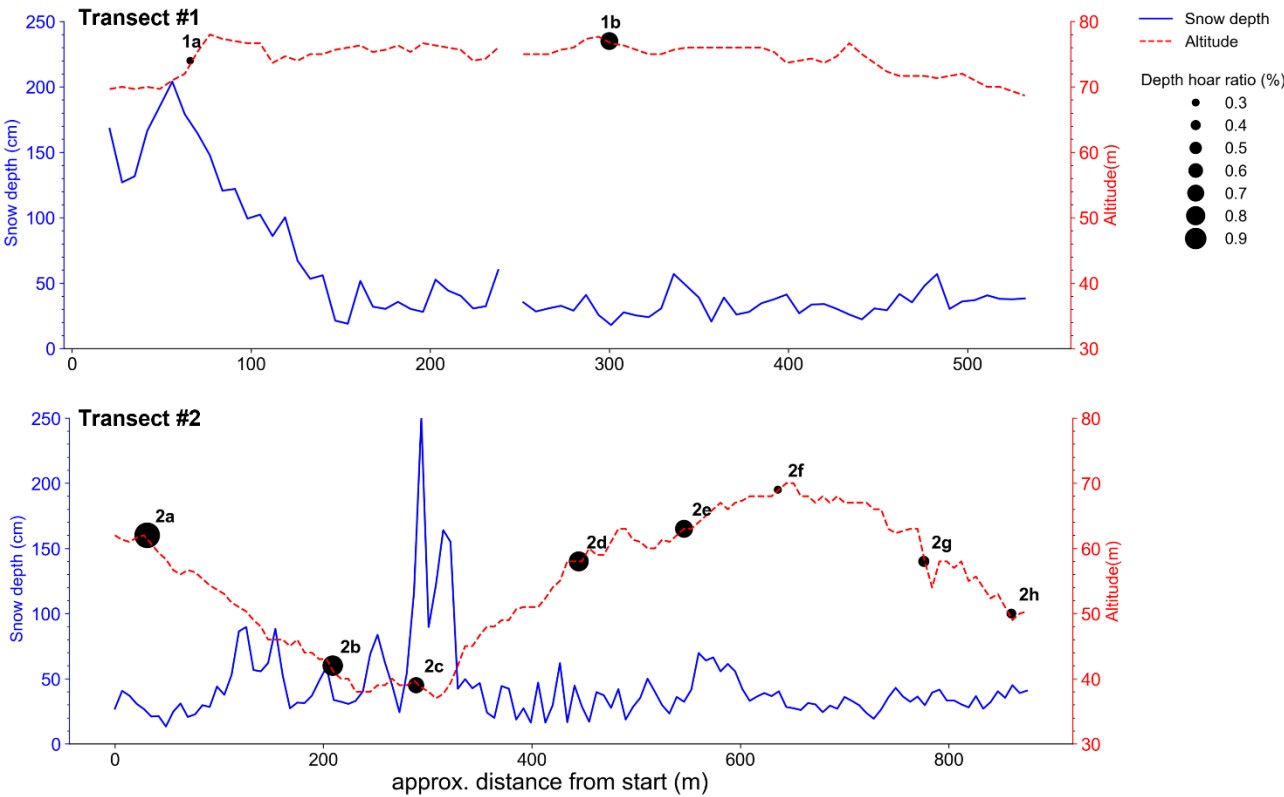


**Figure 4.** Snow depth (SD) transects surveyed in the Ice Creek catchment. Transect #1 and transect #2 shows snow depth (solid line) and altitude (m.a.s.l., dashed line) along transect. Mean Depth hoar (DH) ratio are indicated along transects by proportional size. 1a, 1b and 2c contain one observation. See Fig. 2c for their location in the catchment.



### 4.1.2. Snow characteristics by vegetation classes

*Snow Depth*

The average SD within Ice Creek catchment was 47.4 cm ± 39.6 cm. The range of variability was substantial, with minimum value at 8.0 cm and a maximum at 212.0 cm. The standard deviation of the snow depth was variable yet strong among all classes (Table 3). The largest standard deviation measured was over *Lupine* (22.3 cm or 57 % of the mean SD) followed by *Coltsfoot* (67.6 cm or 54 % of the mean SD). *Coltsfoot* was by far characterized a greater SD than any other class.

Despite the great deviation around the mean for each class, the Welch ANOVA (Table 4) shows that SD is significantly different between all vegetation groups (p-value = $5^{-13}$). Games-Howell post-hoc revealed that *Coltsfoot* SD measurement is significantly different from the other classes as well as *Dryas*. The difference between *Lupine* and *Shrub* is not significant.

*Topographic Wetness Index*

The average TWI was 6.1 ± 1.6. The minimum and the maximum ranged between 2.5 and 14.7 while the average by vegetation

classes range between 7.4 and 5.5 with a weak deviation around the means (Table 3). *Coltsfoot* showed the highest TWI which is consistent with its location in the valley (Fig. 3) and its vegetation group type, characterized by hydrophilic vegetation. The Welch ANOVA shows that wetness index extracted with TWI is significantly different between all vegetation groups with a p-value <0.001 (p-value = $6^{-14}$), which again is expected to lead to different responses from TSX. The Games-Howell post-hoc test revealed that difference in TWI is not significant between *Coltsfoot* and *Shrub* units and between *Dryas* and *Lupine*

(Table 5).

*Depth Hoar Fraction*

The DHF of the snowpack was larger than 0.5 for all vegetation classes. However, standard deviations of classes with shallow snow such as *Lupine* and *Dryas* were greater whereas the standard deviation was lower than 0.1 for *Coltsfoot* where the SD is the highest. At least one horizontal structure (ice layer or melt-freeze crust layer) was found in each snowpit. The average

thickness of each ice or melt-freeze crust layer was 1.6 cm ± 0.7 cm and the cumulative thickness average by snowpit was 4.5 cm ± 2.8 cm. The maximum ice thickness (4 cm) was found at the station downstream, which would suggest that the sensitivity of the CPD to ice layers should be generally lower than what was found by Dedieu et al. (2018). The high stratification otherwise may attenuate the signal.



**Table 3: Averaged SD and DHF for each of the vegetation class**

| Vegetation class | SD and TWI nb of samples | Averaged SD ±σ (cm) | Averaged TWI ±σ | DHF nb of samples | Averaged DHF ±σ (%) |
|---|---|---|---|---|---|
| *Coltsfoot* | 29 | 126.0 ± 67.6 | 7.4 ± 0.9 | 8 | 0.55 ± 0.06 |
| *Dryas* | 146 | 31.8 ± 14.1 | 5.9 ± 1.2 | 16 | 0.62 ± 0.31 |
| *Lupine* | 118 | 38.9 ± 22.3 | 5.5 ± 1.5 | 21 | 0.60 ± 0.21 |
| *Shrub* | 28 | 44.1 ± 20.6 | 6.8 ± 1.3 | 6 | 0.51 ± 0.18 |
| Other units | 50 | 69.3 ± 47.2 | 6.8 ± 2.4 | 7 | 0.58 ± 0.20 |
| **Average in catchment** | **371** | **47.4 ± 39.6** | **6.1 ± 1.6** | **58** | **0.59 ± 0.22** |

**Table 4: Post-hoc analysis with Games-Howell for snow depth in vegetation classes (non-parametric test). Each row present variance between snow depth means from two different groups. All vegetation groups were tested on each other.**

| | | **Snow depth** | | |
|---|---|---|---|---|
| Class 1 | Class 2 | Mean Difference (cm) | Standard error (cm) | p-value |
| *Coltsfoot* | *Dryas* | +94.2 | 12.60 | 0.001 |
| *Coltsfoot* | *Lupine* | +87.1 | 12.71 | 0.001 |
| *Coltsfoot* | *Shrub* | +81.9 | 13.14 | 0.001 |
| *Dryas* | *Lupine* | -7.14 | 2.36 | 0.014 |
| *Dryas* | *Shrub* | -12.34 | 4.07 | 0.015 |
| *Lupine* | *Shrub* | -5.2 | 4.41 | 0.622 |

**Table 5: Post-hoc analysis with Games-Howell for the TWI (non-parametric test). Each row present variance between TWI means from two different groups. All vegetation groups were tested on each other.**

| | | **TWI** | | |
|---|---|---|---|---|
| Group 1 | Group 2 | Mean Difference (Wetness index) | Standard error (cm) | p-value |
| *Coltsfoot* | *Dryas* | +1.56 | 0.20 | 0.001 |
| *Coltsfoot* | *Lupine* | +1.97 | 0.22 | 0.001 |
| *Coltsfoot* | *Shrub* | +0.63 | 0.30 | 0.147 |
| *Dryas* | *Lupine* | +0.41 | 0.17 | 0.076 |
| *Dryas* | *Shrub* | -0.93 | 0.26 | 0.003 |
| *Lupine* | *Shrub* | -1.34 | 0.28 | 0.001 |



## 4.2. TerraSAR-X results

### 4.2.1. Spatial and temporal evolution of CPD

Figures 5a and 5c show the averaged temporal evolution of CPD and CCOH with descending orbit 24 (incidence angle = 31°) for each vegetation class as well as the confidence interval (95%). The period with presence of snow was set between mid-September and mid-May based on prior observations (Burn and Zhang, 2009; Stettner et al., 2018). Figure 5c shows the monthly average temperature and cumulative monthly precipitation on Qikiqtaruk-Herschel Island.

A periodicity was observed with the CPD signal, with on one side, the period of snow-free condition where the signal oscillates around zero, and on the other side, the period with snow where the signal decreased over the season suggesting an influence from the snowpack. For the 2014–2019 period, the mean CPD value during the snow season was -8.59° with annual mean ranging between 13.41° (2014–2015) and -6.42° (2017–2018). During the snow-free condition, the average CPD over the same period increased to -0.87° and ranged between -0.44° (2014–2015) and -1.32° (2015–2016). The decrease generally started in January, when the average air temperature is at its coldest (-20 °C) except in 2016, where a warming occurred, increasing the average temperature of 5 °C for that year. The two classes with taller vegetation type (*Coltsfoot* and *Shrub*) stood out during the winters 2014–2015 and 2016–2017. There the CPD decreased to -60° towards the end of the winter. The decrease in the CPD was therefore similar between the vegetation classes and is about -15°.

Overall, the coherence stayed greater than the 0.5 threshold over 2014–2019 period with an average of 0.71 ± 0.11. The signal was lower during snow-free period where the average is 0.63 ± 0.11. The coherence then increased around 0.76 ± 0.08 during the winter. *Coltsfoot* and *Shrub* classes showed greater variation on the coherence over the seasons and the years compared to *Lupine* and *Dryas* classes. The average CCOH by vegetation classes ranged between 0.69 ± 0.07 (for *Coltsfoot*) and 0.72 ± 0.07 (for *Dryas*).





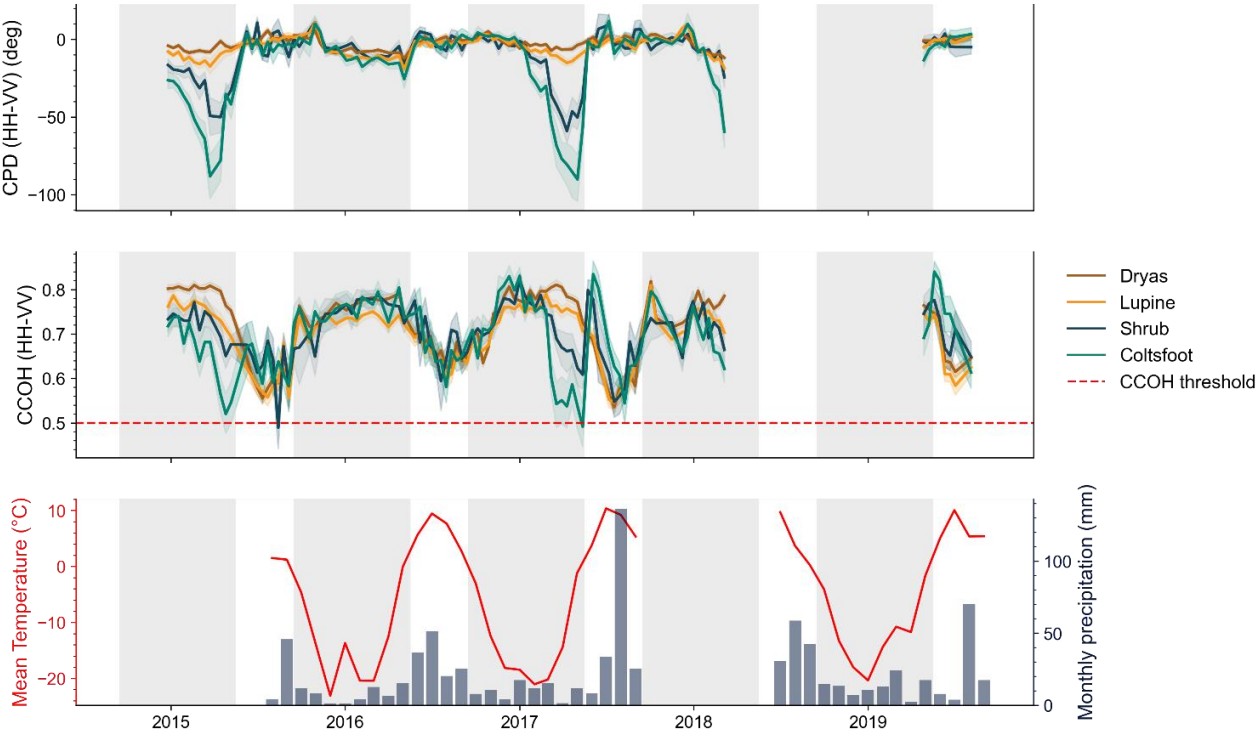


**Figure 5.** (a) Average CPD and (b) Average CCOH by vegetation class with interval of confidence (95%) for orbit 24 (31°, descending). Pixels values were extracted from GPS dataset (see Fig. 2c), where $N_{Coltsfoot} = 33$, $N_{Dryas} = 140$, $N_{Lupine} = 118$, $N_{Shrub} = 29$. Winter period (mid-September to mid-May) is shows in shaded area. Windows pixels size is 1x1 pixel (5x5m). (c) Meteorological data from Qikiqtaruk Herschel Island station (dataset from Environment Canada (2021)). The meteorological station is not equipped with a telemetry system and

since the island is inaccessible during the winter, the lack of data during the winters 2015 and 2018 was caused by a malfunction of the station.

### 4.2.2. Retrieving SD per vegetation class using CPD

The dataset from 2019 was used to perform a simple linear regression analysis allowing to assess whether there is a statistically significant relationship between snow measurements (layer depth of the depth hoar, wind slab, melt-freeze crust and ice layers and mean density of each layer) and CPD. No significant correlation was found other than SD or the samples contained fewer

than 10 observations which bring elusive correlation (see Appendix A for more details). The best correlations between SD and CPD were found with *Lupine* (orbit 24, desc., incidence angle 31°) and *Coltsfoot* (orbit 115, desc., incidence angle 38°) (Table 6).

The *Coltsfoot* and *Shrub* classes were characterized by similar TWI mean values as well as low TWI variance. These two

classes were combined to be compared with other classes (named *Coltsfoot+Shrub* in Table 6). This grouping led to an improvement in the coefficient of determination of 0.044 as well as a decrease in p-value and standard deviation. Samples with a coefficient of determination greater than 0.50 met the assumptions of homoscedasticity as well as the absence of autocorrelation, except for the sample located in *Coltsfoot+Shrub* in the 115 orbit, and samples located in *Coltsfoot* in the





152  asc. orbit (See Table B1 for further details). These results show clearly that CPD can be used to retrieve SD, albeit not in

all vegetation classes.

**Table 6:** R-squared, p-value and standard deviation ($R^2 \pm$ std [$pvalue \pm std$]) results from linear regression analysis between SD and CPD obtained by vegetation classes and by orbits. The confidence interval was measured using the "Boostrap with replacement" resampling technique ($N_{bootstrap} = 1000$). The standard deviation of r-squared and the p-value obtained by the technique are indicated in the results whose variance is explained to more than 50%

|  | *Lupine* | *Dryas* | *Coltsfoot* | *Coltsfoot+Shrub* |
|---|---|---|---|---|
| **Orbit 24 (31°)** | **0.55 ± 0.11** **(0.001 ± 0.004)** | 0.01 *(0.59)* | 0.10 *(0.55)* ** | 0.07 *(0.47)* |
| **Orbit 115 (38°)** | 0.01 *(0.51)* | 0.004 *(0.64)* | **0.72 ± 0.16** **(0.00 ± 0.01)** | **0.74 ± 0.09** **(0.00 ± 0.00)** |
| **Orbit 152 (24°)** | 0.02 *(0.44)* | 0.0 *(0.82)* | **0.68 ± 0.18 ** ** **(0.08 ± 0.08)** | 0.001 (0.91) |

**\*\* Fewer than 10 observations**

## 5. Discussion

### 5.1. Snow distribution on Qikiqtaruk-Herschel Island

*Snow Depth*

On the study site, snow gets quickly redistributed across the landscape by winds. Burn and Zhang (2009) showed that SD distribution patterns were primarily driven by topography in close vicinity to the Ice Creeks. Our observations (Fig. 4) concur

and expand on those from Burn and Zhang (2009) by highlighting the effect of microtopography and of vegetation in controlling SD. The SD was greater (> 100 cm) in areas characterized by shrubs and wetlands (*Coltsfoot*), which are mainly associated with valley bottom locations. There is a significant difference in SD between *Coltsfoot* and any other class, which shows that snow gets blown away on high points and slopes and accumulated in spatially constrained areas at the valley bottom (Fig. 3d and 3c). By contrast, grass-type or shallow vegetation, such as *Dryas* and *Lupine*, is found in wind-exposed areas.

Deeper snow was found over *Lupine* compared to *Dryas*. The microtopography may play a role in this difference, as the standard deviation of SD is greater in the *Lupine*. There is greater variability in SD between the troughs and the top of hummocks, as documented by Wilcox et al. (2019). Thus, we can relate to this study as the distribution of snow in the Ice Creek catchment is driven primarily by vegetation and topography.

*Depth Hoar Fraction*

We suggest that the DHF is strongly driven by microtopography. During winter 2019, the DHF amounted to an average of 59% of the snowpack (n = 58). There is a greater standard deviation of these measurements in vegetation classes where the





average SD is lower (less than 40 cm average depth) such as *Lupine* and *Dryas*. The effect of microtopography allows snow capture in hummock hollows early in the season and the thermal gradient from the ground to the surface varies accordingly (King et al., 2018; Wilcox et al., 2019). Depth hoar develops when a strong thermal gradient occurs between the ground and

the snow surface. There are two situations where a strong gradient occurs: 1) when the SD is low and 2) when the soil is warm and the snow surface is cold. Sturm and Holmgren (1994) have shown that the depressions in tussocks or hummock are warmer than the top. The thermal gradient found in this type of vegetation class may therefore explain the large standard deviation of DHF found in *Lupine* class. Thus, the soil wetness should be higher in the hollows, but that effect might not be captured by the TWI used in this paper as its spatial resolution relies on a 2 m resolution DEM.

*Comparison over Snow Classification*

The snow characteristics observed over the Ice Creek catchment are consistent with the literature. We compared our statistics to the classification proposed by Royer et al. (2021) and observed a good fit with the Herbaceous and low shrub tundra snow class (Table 7). The standard deviation of the mean snow depth from our study site and from Royer et al. (2021) classification are both greater than 80%. The local topography inherited from the last glaciation (Late Wisconsin) is specific to Qikiqtaruk-

Herschel Island and could explain higher snow depth, and therefore higher SWE and density. The maritime effect observed on Qikiqtaruk (Cray and Pollard, 2015) could also explain the warmer mean temperature during winter. All study sites used in Royer et al. (2021) classification are in the East of Canada. Further studies and datasets from the western part of Canada would greatly improve the snow classification.

**Table 7.** Comparison of snow characteristics with Royer et al. (2021) classification. Mean temperature was extracted from 1974-2019
meteorological station from Qikiqtaruk Herschel Island and for the winter season (December to March as define by Royer et al (2021)).

| | Latitude range (°N) | Mean Temperature (°C) | SWE ± Std (mm) | SD ± Std (cm) | Density ±Std (kg m$^{-3}$) |
|---|---|---|---|---|---|
| Qikiqtaruk Herschel Island | 68-69 | -22.1 | 142.6 ± 99.1 | 47.4 ± 39.6 | 343.8 ± 73.7 |
| Herbaceous and low shrub tundra snow (from Royer et al. (2021) | 58-74 | -23.6 | 132.9 ± 57.6 | 43.1 ± 35.2 | 315.3 ± 49.1 |

## 5.2. CPD Spatio-temporal evolution and SD correlation

The high-resolution vegetation classification used in this paper allowed us to show that CPD varies greatly according to seasons and vegetation class (Fig. 5). Overall, the CPD signal decreased during winter and increased rapidly during melt. This concurs with observations from Leinss et al. (2014, 2016) made in Sodankylä. According to the model developed by Leinss (2014,

2016), the strong CPD decrease observed in 2015 and 2017 winters over shrubs areas could be explained by fresh snow





accumulation or dominance in horizontal structures. However, the snow distribution analysis showed that the *Shrub* class has shallow snow, making it, SD-wise, significantly different from the *Coltsfoot* class, meaning the result doesn't show that the measured CPD signal is entirely governed by the snowpack. The CPD evolution over different vegetation classes is significantly different between two distinct groups: tall vegetation zones (*Coltsfoot* and *Shrubs*) and low vegetation zones
(*Lupine* and *Dryas*). The small decrease observed at *Lupine* and *Dryas* classes could indicate an influence from the ground, as the snow depth measured is less than 30 cm and highly stratified. The effect from inhomogeneities within the snowpack does not support this case, as the CCOH is greater of 0.5 for each pixel. Although the snowpack was highly stratified, each ice layers or melt-freeze crust was in average less than 2 cm thick, which is thinner than the ice layers in the snowpacks studied by Dedieu et al. (2018). It may explain why the linear regression analysis of CPD shows the best results with the total SD (*i.e.*
less sensitive to small crusts), which has never been observed before.

A high level of moisture in the ground will lead to major dielectric contrast at the snow-soil interface, hence limiting the penetration depth of the radar signal (Duguay et al., 2015). Thus, the sensitivity of the signal to ground conditions decreases. Duguay et al. (2015) also showed a strong saturation of TSX signal in the areas with shrubs greater than 50 cm. The shrubs may explain the best correlation observed in Table 6 as Myers-Smith et al. (2019) report an increase of the canopy where the
measured shrubs at the bottom of the valley were more than a meter.

The TWI variance analysis shows that there is no significant variance between *Coltsfoot—Shrub* classes and between *Lupine—Dryas* classes, which could explain the strong decrease of the signal observed in mid-winter (Fig. 5). A high TWI indicates a high-water accumulation potential, hence a higher saturation of the soil. In the microwave range, soil saturation increases the dielectric properties of the soil. The sensitivity of the X-band radar signal is then higher, which allows the interface between
the snowpack and the ground to be well discriminated. Thus, CPD captures snow accumulation well across winter in areas of higher potential of soil moisture, while soils with lower potential moisture are likely to contribute to the CPD signal and thus reduce the correlation between snow depth and CPD signal.

We suggest the increase in the r-squared depends on the soil moisture because there is less contribution from the ground on the backscatter signal. A higher incidence angle (> 30°) improve the results, agreeing with Leinss et al. (2014). The use of the
TWI is promising for the snow-SAR dynamics as it is easy to compute and relies on topographic datasets that are now widely available for the entire Arctic. Furthermore, no correlation was found between retrieval performance of SD and DHF, suggesting that the poor performance over *Dryas* class is explained by soil contributions. The relatively conclusive results for the *Lupine* class at orbit 24 show an inverse correlation (Fig. 6) which contradicts with Leinss et al. (2014). We hypothesize that the tussock depressions are preferential areas for the formation of depth hoar, caused by the effect of microtopography.
Thus, vertical structures are dominant in the snowpack, which could explain an increase of vertical structure where the snowpack is deeper at this vegetation class. Further analysis should be done on the soil moisture and on the effect of the depth hoar distribution to better capture the wetness of hummocky area and how it can improve retrievals of SD.





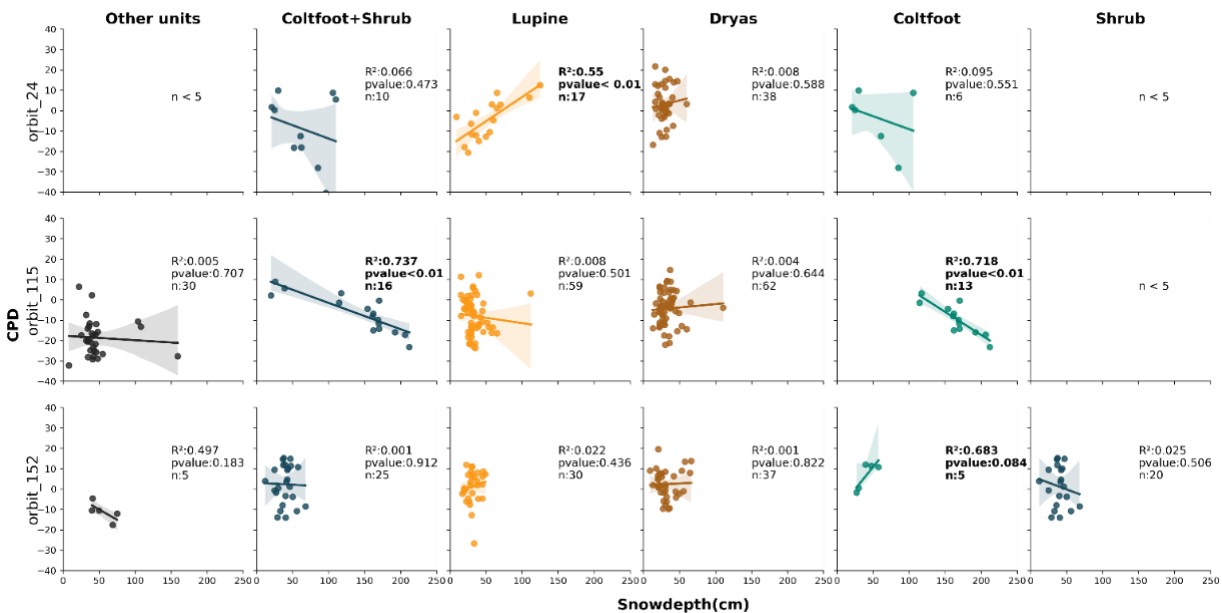

**Figure 6.** Correlation analysis between CPD and snow depth.

## 6. Conclusion

This study was the first to investigate the potential of co-polar phase difference (CPD) derived from TerraSAR-X data in combination with snowpit characterization over Qikiqtaruk-Herschel Island. We were able to find a variability in SD and TWI depending on vegetation classes extracted from a high-resolution map of vegetation cover. Classifying snowpits by vegetation classes on Qikiqtaruk-Herschel Island shows respectable results, helping to demonstrate the effect of topography and hence the moisture rate of the ground on CPD signal. The 2019 dataset shows a high heterogenous snowpack with different ice layers and with a DHF representing in average more than half of the snowpack.

Despite this complex snowpack, we demonstrate a correlation between the CPD and the SD when certain conditions are met. A high incidence angle (> 30°) with a high TWI (> 7.0) allows to show a correlation between SD and CPD with R-squared up to 0.72. CPD cannot be used to extract the fresh snow in an arctic context, as the penetration of the electromagnetic wave tend to go through the entire snowpack.

The *in-situ* data used for the present study do not cover the entire winter on Qikiqtaruk-Herschel Island, which brings uncertainties on snow depth characterization with CPD. The maritime climate of Qikiqtaruk-Herschel Island may advance the snow melt period and provoke a shift to a wet metamorphism regime of the snowpack. The lack of consistent stratigraphy measurement over the winter is still a major limit in snow studies. Consistent stratigraphy measurement over the winter would improve the understanding of the snowpack metamorphism regime.



Focus of future studies could be the threshold sensitivity to TWI and the incidence angle of snow depth retrievals to map snow depth in such environments and to evaluate the potential of using interpolation tools to cover the gaps in SD information over dryer vegetation types.

*Code and data availability.* Data and code are made available upon request from the corresponding author.

*Author contributions.* JVS performed this study as part of her Master thesis project, co-supervised by AL and HL. JVS, SS, and AW designed the methodology. JVS wrote the code, performed the field measurements and the analysis. The original idea and method were developed by JVS, AL and HL. AW performed the SAR preprocessing, AR, AW and JPD support in the
SAR analysis. The manuscript was written by JVS and edited by all the co-authors.

*Competing interests.* The authors declare they have no conflict of interest.

*Acknowledgment.* This work was funded by Mitacs-Globalink, the EU Horizon 2020 program (Nunataryuk, grant no°773421
and Polar Knowledge Canada. TerraSAR-X imagery was provided by the German Aerospace Centre (© DLR 2019). We would like to thank the Aurora Research Institute, the Yukon Territorial Government and Yukon Parks (Herschel Island Qikiqtaruk Territorial Park) for administrational and logistical support, and the Inuvialuit people for the opportunity to conduct research on their traditional lands. JVS would like to thank Vincent Sasseville for the fieldwork support and Silvan Leinss for the several exchanges and discussion on CPD method.

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






## Appendix A: Complementary results

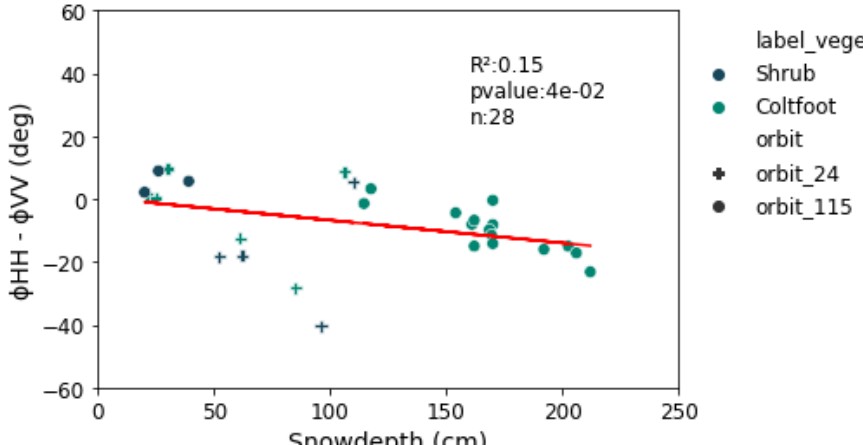

**Figure 1A:** Linear regression between CPD and snowdepth on descending orbit (orbits 24 and 115) without TWI threshold**.**


Table A1 show the complementary results retrieved during the linear regression analysis, between CPD and every snow variable measured on the field. Results with $R^2$ greater or equal to 0.5 are shown in bold character. The samples may vary when a measurement was not possible during the field campaign.

Each variables describe a characteristic sampled in the snowpit:

H_tot : Snow depth (cm).

H_ws : Wind slab height (cm).

H_dh : Depth hoar height (cm).

H_tot_ice : Snow height to the first ice layer observed in the snowpit (cm). The ice layer thickness must be greater than 2 cm.

H_tot_mf : Snow height to the first meltfreeze crust (cm).

Density_moy : Average snow density (kg m-3)

Density_dh : Average density for the depth hoar layer (kg m-3).

Ssa_ws : Average *snow surface area* measured in the wind slab layer.

Ratio_df : Depth hoar fraction in the snowpit (%)

Cumul_tot : Cumulative thickness of horizontal layers (meltfreeze crust, ice lens, in cm).

Ice_cumul : Cumulative thickness of ice lens (cm). cumulée de couche de glace

Cumul_crust : Cumulative thickness of meltfreeze crust (cm).

Ratio_ws : Windslab ratio in the snowpit (%)



**Table A1.** Complementary results retrieved during the linear regression analysis. The standard deviation with bootstrap is not show as the results with $R^2$ greater than 0.5 have samples with less than 8 observations.

### Orbit 115

| | ratio_dh | ice cumul | cumul crust | density moy | density dh | ratio_ws | h_tot_mf | h_tot_ice | h_dh | h_ws | h_tot | ssa_ws |
|---|---|---|---|---|---|---|---|---|---|---|---|---|
| **$R^2$ (p)** | 0.0 (0.86) | 0.16 (0.15) | 0.17 (0.13) | 0.06 (0.42) | 0.03 (0.60) | 0.00 (0.92) | 0.00 (0.93) | 0.07 (0.36) | 0.22 (0.08) | 0.09 (0.28) | 0.31 (0.03) | 0.05 (0.46) |
| **sample** | 15 | 15 | 15 | 14 | 13 | 15 | 15 | 15 | 15 | 15 | 15 | 13 |

### Orbit 152

| | ssa_ws | ice_cumul | **cumul tot** | h_tot | **h_ws** | h_dh | h_tot mf | h_tot ice | ratio_ws | density dh | **density moy** | cumul crust | ratio dh |
|---|---|---|---|---|---|---|---|---|---|---|---|---|---|
| **$R^2$ (p)** | 0.04 (0.7) | 0.46( 0.06) | **0.64 (0.02)** | 0.45( 0.07) | **0.48 (0.06)** | 0.02 (0.72) | 0.38 (0.19) | 0.19 (0.28) | 0.09 (0.56) | 0.0 (0.98) | **0.51 (0.05)** | 0.34 (0.17) | 0.27 (0.19) |
| **sample** | 6 | 8 | 8 | 8 | 8 | 8 | 6 | 8 | 6 | 8 | 8 | 7 | 8 |

### Orbit 24

| | cumul crust | ratio ws | density moy | density dh | ratio dh | ssa ws | h_tot_ice | h_dh | h_ws | h_tot | ice_cumul | h_tot_mf | cumul_tot |
|---|---|---|---|---|---|---|---|---|---|---|---|---|---|
| **$R^2$ (p)** | 0.02 (0.64) | 0.02 (0.6) | 0.01 (0.7) | 0.09 (0.33) | 0.0 (0.83) | 0.01 (0.76) | 0.03 (0.59) | 0.04 (0.49) | 0.03 (0.59) | 0.0 (0.87) | 0.2 (0.14) | 0.03 (0.55) | 0.08 (0.33) |
| **sample** | 13 | 14 | 14 | 13 | 14 | 12 | 12 | 14 | 14 | 14 | 12 | 13 | 14 |



## Appendix B: Statistical analysis assumptions and results

**Homoscedasticity**

Linear least squares regression assumes that the residuals come from a population where the variance is constant. When heteroscedasticity is present, the result is therefore unreliable. The Breusch-Pagan statistical test evaluates the assumption of homoscedasticity, *i.e.* the consistency of the error variance in a linear regression model.

The assumptions are:

*null hypothesis (H0):* Homoscedasticity is present

*alternative hypothesis (Ha):* Homoscedasticity is not present (heteroscedasticity is present)

If the p-value of the Lagrange multiplier statistic (LMS) is greater than 0.05, the probability of homoscedasticity is greater than 5%. The null hypothesis is therefore retained. In the opposite case (p-value $< 0.05$), the probability of homoscedasticity is less than 5%. The alternative hypothesis is then adopted.

**Autocorrelation**

The Durbin-Watson test (DW) is used to test the autocorrelation of residuals in linear regression models. It assesses whether the residuals are independent.

The assumptions are:

*H0:* There is no correlation between the residuals

*Ha:* The residuals are autocorrelated.

The results are expected between 0 and 4. Values between 1.5 and 2.5 indicates no autocorrelation. Results near 0 show positive autocorrelation, while results near 4 show negative autocorrelation.





**Table B1.** statistical test results for CPD and snowdepth correlation analysis. Each model represents a divided sample in function of vegetation class and TSX orbit. Results with an $R^2$ greater than 0.5 is shown. Durbin-Watson test (DW) and the Breusch-Pagan test (LMS) were selected to assess the autocorrelation for the first and the homoscedasticity for the latter. LMS stand for Lagrange multiplier statistic

| Vegetation class | orbit | sample | $R^2$ | Adjusted $R^2$ | DW | LMS | LMS p-value |
|---|---|---|---|---|---|---|---|
| *Lupine* | 24 | 17 | 0.55 | 0.52 | 1.80 | 2.51 | 0.11 |
| *Coltsfoot* | 115 | 16 | 0.70 | 0.68 | 2.01 | 1.65 | 0.19 |
| *Coltsfoot+Shrub* | 115 | 19 | 0.75 | 0.73 | 0.81 | 3.48 | 0.06 |
| *Coltsfoot* | 152 | 5 | 0.68 | 0.58 | 1.37 | 2.03 | 0.15 |