# Peer review of "Potential of X-band polarimetric SAR co-polar phase difference for Arctic snow depth estimation"

_The Cryosphere, 2021_

## Author Comment (AC1)

**Potential of X-band polarimetric SAR co-polar phase difference for Arctic snow depth estimation**
**tc-2021-314**

**Revisions**

Joëlle Voglimacci-Stephanopoli, Anna Wendleder, Hugues Lantuit, Alexandre Langlois, Samuel Stettner, Jean-Pierre Dedieu, Achim Roth, and Alain Royer

Community member's comments
Answers to reviewer
***Modification to text***

Dear Georg,

Thank you very much for the useful comments regarding our work and especially all the suggestions regarding the polarimetry. They helped a lot to improve the manuscript. We deeply appreciate your conscientious review. For a better readability of our response, the answers to the reviews and the corrections in the manuscript are shown in orange.

**CC comments 1:** It's clear that you did a lot of analyses on a lot of data, but I found it sometimes difficult to follow all the analyses across different measurements and vegetation classes. For instance, when do you speak about the long 5-year TSX time series and when about the short interval with the 3 orbits? You write that you have snow measurements which are revisited for each TSX acquisition, but that seems to apply only to the 2019 data.

We agree this structure might be difficult to follow. In order to improve readability, we modified the objectives and add a third one (modification at lines 64-67):

[…] (1) investigate SD and DHF variability between different vegetation classes in the Ice Creek catchment (Qikiqtaruk-Herschel Island, Yukon, Canada) using *in situ* measurements collected during a field campaign in 2019; ***(2) evaluate linkages between snow characteristics and CPD distribution over 2019 dataset and (3) Determine CPD seasonality considering meteorological data over 2015-2019 period.***

We mentioned the period of acquisition for the snow profiles (line 150) and their utility (line 158). We also suggest a modification for the sentence at line 158 to improve the understanding of the reader:

"Detailed snow profiles were acquired in spring 2019 (mid-April to early May)".

"The snow depth and mean density of each layer classified were compiled in a linear regression analysis with TSX data ***from the same period. Regression analysis will be used to reach objective (2) of this paper.***

We took the 5 years TSX timeseries as it allowed reporting on the seasonality of CPD signal over the years. Unfortunately, no snow information was available starting 2015 nonetheless, the objectives stated were reached.

**CC comments 2:** A thorough check, which results are important and which could be

removed for conciseness, could also help the reader and better tailor the paper towards the objective(s). For instance, in section 4.2.1, the reported seasonal values for CPD are very important, but annual means might be not so meaningful with such distinct snow and snow-free seasons. Similarly, I'm not sure how the paragraph on "Comparison over Snow Classification" contributes to the objective of the paper (but this impression could be just because of my radar perspective).

Sentences on line 303-305 were meant to report seasonal values over each year. To improve clarity the text was changed to:

"For the 2015-2019 period, the mean CPD value during the snow season was -8.59**°. *The means of each winter are ranging between 13.41° (2014-2015) and -6.42° (2017-2018).* During the snow-free condition, the average CPD over the same period increased to -0.87° (2015-2019**). *Maximum and minimum values during snow-free conditions ranged between -0.44° (2015) and -1.32° (2015-2016). "***

Regarding the paragraph "Comparison over snow classification", we have the feeling that this paragraph is necessary to contextualize our dataset to the newly updated snow types classification proposed by recent work from our group by Royer et al. (2021) following results from Sturm (1995) which demonstrates the applicability of arctic snow classifications in Western Canada. As such, to improve the flow of the manuscript, we suggest moving this paragraph to the Conclusion section and moving Table 2 in appendix.

**CC comments 3:** The analyses of the wealth of in situ data and co-polar phase measurements is for sure highly valuable for the scientific community, but I'm wondering if the derived conclusions could be clearer and more elaborate. I think 3.5 lines of conclusions about CPD and snow depth could be a bit more when looking at the title of the paper. For instance, conclusions about which scenarios do not give a correlation between snow depth and CPD measurements and the underlying reasons (some ideas: shallow snow at exposed topography, maybe related to certain snow structures like wind crusts or depth hoar, which don't have the required anisotropy to give CPD. Generally small sensitivity to shallow snow. Certain ground conditions, even though I don't understand what you mean there, see questions below). Maybe also some thoughts about how to overcome these limitations could be of interest.

Thank you for the suggestion. We included a paragraph 'Future work' and detailed some thoughts:

"Future studies should focus on the threshold sensitivity to TWI and the incidence angle of snow depth retrievals in order to map snow depth in such environments. This would also allow an evaluation of the potential of using interpolation techniques to bridge spatial observational gaps in SD information at the watershed scale**. *First, SD variability within a TSX pixel should be studied further, especially in hummocky areas where the highest variability was found, which could suggest a variability in the TWI as well. Statistical approaches, using the coefficient of variation of snow depths (CVsd), as suggested by Winstral and Marks (2014) and Liston (2002) could be an interesting avenue in the development of a representative mapping of the terrain. Meloche et al. (2022) demonstrated recently the effectiveness of the coefficient of variability of snow depth (CVsd) to improve passive microwave SWE retrievals in similar environment found on Herschel island (i.e., arctic snowpack with tundra vegetation type).***

References added:

Liston, G. E.: Representing Subgrid Snow Cover Heterogeneities in Regional and Global Models, Journal of Climate, 17, 1381-1397, 10.1175/1520-8, 2014. 0442(2004)017<1381:RSSCHI>2.0.CO;2, 2004.

Meloche, J., Langlois, A., Rutter, N., Royer, A., King, J., Walker, B., Marsh, P., and Wilcox, E. J.: Characterizing tundra snow sub-pixel variability to improve brightness temperature estimation in satellite SWE retrievals, The Cryosphere, 16, 87–101, https://doi.org/10.5194/tc-16-87-2022, 2022.

Winstral, A., and Marks, D.: Long-term snow distribution observations in a mountain catchment: Assessing variability, time stability, and the representativeness of an index site,13 Water Resources Research, 50, 293-305, 10.1002/2012WR013038, 2014.

**CC comments 4:** The correlation between CPD and SD shown in Table 6 gives higher correlation for some vegetation classes, while the correlation results in Appendix A give only low correlation for SD (H_tot). Do I understand it right that this is because all vegetation classes are combined in Appendix A? And how does this relate to line 325 "No significant correlation was found other than SD.." ?

Yes, the understanding is correct. Our samples contain 15 observations or less for each snow characteristics, which we feel is to divide the sample by vegetation classes. Appendix A (Changed to appendix B) shows no significant correlations in the snowpack characteristics at each orbit. Only 2 correlations are possible at orbit 152 (incident angle=24°, on the cumulative thickness of horizontal layers (meltfreeze crust, ice lens, in cm) and mean density of the snowpack) but they are not significant because of the sample size (8 observations for each characteristics).

**CC comments 5:** The discussion about TWI is interesting, but beyond the potential difference in soil moisture, isn't also the question of freezing of soil relevant? In my understanding, any level of soil moisture will give surface scattering from the ground below the snow, which is the desired scattering scenario for the CPD model. Isn't the question rather what happens when the soil freezes?

High moisture in the soil will have the effect to delay the freezing process at first, and then keep the ground temperature stable longer than soil with low moisture (e.g.: Romanovsky and Osterkamp, 2000). On the other hand, Burn and Zhang observed a delay on active layer freeze back in area where "snow may accumulate in early winter" (from section 5.5. of their paper). Active layer in these areas freeze back in mid-December, or a month later than other location (between 2003-2007).

Considering these two different scattering mechanisms:
1) dry snow over wet soil: the SAR signal penetrates through the snowpack and is reflected away by the wet soil. Based on the Romanovsky and Osterkamp (2000) theory and in situ observations from Burn and Zhang (2009), we could suggest that could enhance surface scattering processes of specular reflectance depending on surface roughness. This could explain why stronger correlations with snow depth are observed in area with high TWI, no matter the snow depth as good correlations were observed in Coltsfoot (mean SD: 126.0 ± 67.6) and Lupine areas (mean SD: 38.9 ± 22.3).

2) dry snow over ice: the SAR Signal penetrates the snowpack and scatters on the ice layer. As detailed on lines 390-393, Dedieu and al. (2018) monitored the phenomenon that the SAR signal is not able to penetrate ice layers thicker than 5 cm. The scattering

mechanism on ice is mostly specular reflectance given the flat nature of ice layers. Hence, both an ice layer and a wet soil supports the CPD measurement.

In our case, ice layers in the snowpack were less than 2 cm. It is possible that, in preferential area for water accumulation, ice layers developed at the snow-ground interface which would enhance the surface scattering in the season i.e. after January as observed in figure 5, where the active layer should be frozen. Unfortunately, this was not documented on the 2019 field campaign. A more "in depth" study on freeze up process including in situ data and observations at different vegetation classes would be of great interest to have a better understanding of the processes in place during changes in the CPD signal throughout the season as observed on figure 5.

Reference cited:
Romanovsky, E. and Osterkamp, E.: Effects of Unfrozen Water on Heat and Mass Transport Processes in the Active Layer and Permafrost, 2000.

**CC comments 6:** In the discussion and conclusion about TWI, there is potentially an unclear causality. Maybe the good correlation between CPD and SD for high TWI is rather related to the fact that high TWI values are found in the depression areas which are naturally with high SD (and are apparently the Coltsfoot class)? Similarly, the good correlation between CPD and SD for Coltsfoot could be just because Coltsfoot is predominantly in valleys. I'm wondering if the larger SD (in valleys with coltsfoot) is required to have a certain sensitivity of CPD to SD and the high TWI and related soil moisture is just a correlation but not the cause of the CPD to SD correlation.

We agree that future works should study more precisely the snow depth variability within a vegetation class, as its location is dependant to the topography. We found the best results of snow depth and CPD in Coltsfoot valley, but also in Lupine class, where the mean snow depth is 38.9 ± 22.3 cm which is more than 80 cm different from the mean snow depth in Coltsfoot class (see Table 3). Although there is no certainty that there is no causality between TWI and snow depth, our results show good correlation between a variety of snow depth. Future works should address on the variability of snow depth within a SAR pixel by vegetation class would greatly improve our comprehension at this point. This topic is now addressed in the conclusion. Please refer to our answer above in comment #3.

**CC comments 7:** Line 230: What do you mean by the presence of ice leads to better reflection conditions for the microwave? Do you consider the mentioned moisture content to be frozen or liquid? As you mention somewhere else, larger moisture gives higher dielectric contrast and thus more backscatter, therefore I'm not sure how ice (with less dielectric contrast) leads to a better reflection. And do you maybe mean backscatter instead of reflection here? (forward reflection would reduce backscatter for a side looking SAR)

The presence of an ice layer in the snowpack simply provides a nice surface for specular reflection of the radar signal, especially in the horizontal polarization thus reducing backscatter. To improve clarity the sentence was changed to:

**_"High moisture content at the soil surface would potentially improve the performance of SD retrieval, given that the penetration of the signal into the soil would be limited by the high dielectric constant of the soil."_**

**CC comment 8:** A few statements about the scattering scenario (scattering only from ground) are unclear to me:
Line 424: Isn't the signal penetration through the entire snowpack and only scattering from the ground exactly the desired scenario for the approach of Leinss et al?

Leinss et al.(2014) showed a correlation with cumulation of fresh snow. In our case, our correlation is with the total snow depth, which is a complex snowpack with a diversity of snow type (depth hoar, ice lenses, wind slabs, etc.)
Please refer to lines 109-113 form the manuscript:

"A relationship was found between CPD and snowfall by Chang et al. (1996) and Leinss et al. (2014) which induces a propagation delay among horizontal and vertical phases due to horizontal alignments of fresh snow crystals. Recent studies focused on the boreal region (Leinss et al., 2014, 2016) or were applied in arctic region with no or sparse vegetation (Dedieu et al., 2018) so the application of the CPD method in the Arctic remains poorly documented. It could be hypothesized that the CPD can describe the entire snowpack in such cold and dry environments."

**CC comments 9:** Line 403: In my understanding, the approach requires all backscattering to come from the ground, which reads the opposite in this sentence? I'm not sure if you mean the contribution of the ground on the backscattered intensity or the CPD by "backscatter signal". It seems to me that you indicate at some occasions that scattering from the ground at low moisture could influence the CPD, but I couldn't find an explanation why. Leinss et al., 2014 mention that potential CPD contributions from rough surface scattering from the soil are small and have the opposite sign. Is this meant here?

The reviewer is right, this is simply a wording mistake, so the sentence was modified as followed:

"Thus, CPD captures snow accumulation well across winter in areas of higher potential of soil moisture, while soils with lower potential moisture are likely to contribute much less to the CPD signal, thus reducing the correlation between snow depth and CPD."

**CC comments 10:** Line 385: What do you mean by "the small CPD decrease during winter for Lupine and Dryas indicates an influence from the ground" if the general scenario is that all scattering comes from the ground anyway? I miss an explanation how ground can influence the CPD measurement.

Our results suggest that high soil moisture would enhance the contrast between ground and snow. Therefore, a low soil moisture would probably lead to signal noise at the interface from vegetation and ground variability.

Sentence in lines 385-388 is modified accordingly:

"The small decrease observed at Lupine and Dryas classes *during the snow season (Fig. 5)* could indicate an influence from the ground, as the snow depth measured is less than 30 cm and highly stratified. *However,* the effect from inhomogeneities within the snowpack does not support this case, as the CCOH is greater of 0.5 for each pixel.
*Dryas is characterized by the lowest TWI, which could lead to less backscattering at the snow-ground interface hence decrease the change in the snow season. High DHF in Lupine vegetation class indicate a potential of higher TWI in the tussoc's hollow, which might not be captured by the TWI. Hence, the TWI variability within a TSX pixel at this vegetation class area could also explain the*

*low decrease of CPD observed in Fig. 5.*

**CC comments 11:** Line 393: I don't understand why "the shrubs may explain the best correlation" and how this is related to the canopy and the size of the shrubs.

Sorry this was a shortcut. Warmer ground temperature was measured under shrubs (e.g.: Domine et al. (2016) and Myers-Smith and Hik (2013)) which could delay the freezing process during the shoulder season (as suggested and detailed on comment #5).
line 393 is changed for the following sentence:

"A high level of moisture in the ground will lead to major dielectric contrast at the snow-soil interface, hence limiting the penetration depth of the radar signal (Duguay et al., 2015). Thus, the sensitivity of the signal to ground conditions decreases. Duguay et al. (2015) also showed a strong saturation of TSX signal in the areas with shrubs greater than 50 cm. ***Warmer ground temperature were previously observed in permafrost are (e.g. Myers-Smith and Hik (2013), Domine et al. (2016). which could delay the freezing process and enhance the contrast at the snow/ground interface. In the case of the study area,*** Myers-Smith et al. (2019) report an increase of the canopy where the measured shrubs at the bottom of the valley were more than a meter.

Reference added :
Myers-Smith, I. H., & Hik, D. S. Shrub canopies influence soil temperatures but not nutrient dynamics: An experimental test of tundra snow-shrub interactions. Ecology and Evolution, 3(11), 3683–3700. https://doi.org/10.1002/ece3.710, 2013.

**CC comments 12:** Retrieving HHVV* phase with arctan() of the Kennaugh elements is only partly correct. Values above +pi/2 and below -pi/2 give a phase jump that causes ambiguous values. I assume the correct functionality of typical programming languages is used to derive the phase, but it might be worth checking to avoid related errors. For the equation, the angle symbol ∠ comes to mind here instead of the arctan().

**Answer 12:** You are right and thank you for your notice. We have used the atan function in idl using two arguments. Hence, the result is between -pi and pi and ambiguous values were intercepted. To avoid misunderstanding, we changed the equation as follows:

$$\phi_{HH} - \phi_{VV} = \angle \frac{K_7}{-K_3}$$

**CC comments 13:** The use of the Kennaugh elements could be clearer. The conventional notation of the full-pol Kennaugh matrix follows K_11 ... K_44, even though I see the single digit notation, with e.g. K_7, in Schmitt et al., 2015. Furthermore, eqs. (2) and (3) are for a Dual-pol Kennaugh matrix, which might confuse readers who are familiar with the conventional Kennaugh matrix. Maybe I am just not aware of this kind of formulation in other literature, but for me it is a particular dual-pol Kennaugh formulation. This can be easily solved by just explicitly stating that you follow the dual-pol Kennaugh matrix formulation of Schmitt et al., 2015, but it might be also related to my finite overview of Kennaugh matrix theory.

Thanks for your comment. To avoid misunderstandings, we will point out to the reader that we have used the Kennaugh notation according to Schmitt et al. (2015).
Therefore, we added following explanation (line 211):

*"Note that the notation of the Kennaugh matrix is labelled according to Schmitt et al. (2015)."*

**CC comments 14:** A detail on eq. (4): The expression of the HHVV* coherence in terms of Kennaugh elements only gives the real valued coherence magnitude, but the left hand term is the complex co-pol coherence (with the CPD phase). Maybe this is meant by the approximate equal sign, though. You could appropriately extend the Kennaugh expression with eq. (1) to integrate the phase and make also the Kennaugh right hand term complex. Alternatively, you remove the CPD phase term and make the entire equation real-valued and only about the coherence magnitude. This would then fit to the way you describe and use CCOH.

That's correct, we lost the complex term on the right part of the formula. We corrected the equation as followed:

$$\gamma_{VV,HH} \cdot e^{i\phi_{CPD}^{\gamma}} = \frac{\langle S_{VV} \cdot S_{HH}^* \rangle}{\sqrt{\langle |S_{VV}|^2 \rangle \cdot \langle |S_{HH}|^2 \rangle}} \approx 2\sqrt{\frac{K_3^2 + K_7^2}{K_0^2 - K_4^2}} \cdot e^{i \angle \frac{K_7}{-K_3}}$$

**CC comments 15:** Eq. (4): Since you use HH-VV phase, meaning negative values for fresh snow, see eq. (1), I suggest to switch the VV and HH subscripts in eq. (4), to make the sign of the CPD phase consistent with eq. (1).

Suggestion accepted

$$\gamma_{HH,VV} \cdot e^{i\phi_{CPD}^{\gamma}} = \frac{\langle S_{HH} \cdot S_{VV}^* \rangle}{\sqrt{\langle |S_{HH}|^2 \rangle \cdot \langle |S_{VV}|^2 \rangle}} \approx 2\sqrt{\frac{K_3^2 + K_7^2}{K_0^2 - K_4^2}} \cdot e^{i \angle \frac{K_7}{K_3}}$$

**CC comments 16:** Line 297 should start "Figures 5a and 5b", I guess.

Thank you. Suggested change was done

---

## Author Comment (AC2)

**Potential of X-band polarimetric SAR co-polar phase difference for Arctic snow depth estimation**
**tc-2021-314**

**Revisions**

Joëlle Voglimacci-Stephanopoli, Anna Wendleder, Hugues Lantuit, Alexandre Langlois, Samuel Stettner, Andreas Schmitt, Jean-Pierre Dedieu, Achim Roth, and Alain Royer

Reviewer's comments
Answers to reviewer
***Modification to text***

Dear Reviewer,

Thank you very much for the careful review and suggestions for improving the manuscript. They helped a lot to improve the manuscript. For a better readability of our response, the corrections in the manuscript are shown in orange.

**Reviewer comment 1:** It did take several reads to parse the stats component of the results section – I think it could be explained a bit more clearly. e.g. what exactly is the post-hoc Games-Howell testing for beyond the non-parametric ANOVA? A sentence to explicitly state why you are using them both would be helpful. What does it mean when the ANOVA says there are no differences, but the Games-Howell also reports only some significant differences? (lines 270-273) If you are reporting it to show the overall group stats as a whole, and then breaking them down further using the Games-Howell that would be useful to point out.

Thanks a lot for the comments. The ANOVA is used to test the variance between the groups. We have chosen Welch's ANOVA as it is a non-parametric test without assumptions of equal variances in the groups. ANOVA gives an overall result regarding the statistically significant differences between the groups. Hence, a post-hoc test allowed us to identify significant differences between specific groups.

For a better understanding, we added following explanation (line 245-246):

"The Shapiro-Wilk test was used to test the normality of distributions for SD and TWI. Since TWI and SD distributions did not respect a normal distribution, the variance in TWI and SD between each group was tested with the non-parametric test Welch ANOVA in conjunction with a post-hoc Games-Howell test. ***Welch's ANOVA allows testing at first if the differences between the groups is statistically significant, while the post-hoc Games-Howell test highlights the differences between specific groups. It may be possible that some groups show no statistically significant difference of the means. For instance, we could expect no difference of the means on the SD and TWI between the group Coltsfoot and Shrubs as both vegetation units are located in areas well suited for snow and water accumulation.*** "

**Reviewer comment 2:** How should it be interpreted when results contradict – e.g. the

TWI results contradict significance between the 2 tests run. This could all be my misinterpretation of the writing, but in that case, some clarification will make this much clearer.

In the case of TWI, we found no statistical significance between the groups "Coltsfoot" and "Shrubs" and between "Dryas" and "Lupine" which indicates that no differences in the potential of water accumulation between these groups was observed. These observations are discussed according to their impact on SAR observation in p.18 (lines 396-402).

**Reviewer comment 3:** Section 3.3 snow-SAR correlation – consider revising this wording as you did not do any correlations, you did regressions. Perhaps choose an alternate word like relationship? Or fit, since you did coefficient of determination. Some lines of text also mix correlation with regression (e.g. 325: No significant correlation was found), technicality perhaps, but you didn't test for correlation.

We changed the section title to "**3.3 Snow - SAR relationship**"

**Reviewer comment 4:** Table 2 raised more questions for me than it answered. It is unclear which in situ dates relate to which acquisition. The text says they are within plus/minus 2 days of the acquisition, but what dates are the actual acquisition from?

The reviewer is right, it is not clear which TerraSAR-X acquisitions were used for the linear regression with the snow pit data. We selected the TerraSAR-X data acquired before and after (± 2 days) of the snow pit measurements. In order to avoid confusion, we added the acquisition dates of TerraSAR-X data used for the linear regression in parentheses. Dates with snow acquisitions (i.e. TSX passing ± 2 days) are listed under "In situ data" row and the number of scenes per orbit is now the quantity if scenes used for the paper in Table 2:

| Relative orbit | Flight direction | Polarization | Incidence angle | Observation period (yyyy-mm-dd) (Acquisition date used for linear regression) | Number of scenes | *In situ* data |
|---|---|---|---|---|---|---|
| 24 | Descending | HH, VV | 31° | 2014.12.26—2018.03.06 2019.04.17—2019.05.20 (2019.04.17, 2019.04.28) | 104 | 2019.04.18 |
| 152 | Ascending | HH, VV | 24° | 2019.04.15—2019.05.18 (2019.04.26) | 1 | 2019.04.26 |
| 115 | Descending | HH, VV | 38° | 2019.04.23—2019.05.15 (2019.04.23, 2019.05.04) | 3 | 2019.04.22 2019.05.03 2019.05.04 |

Modification in the manuscript are done (lines 190-192):

***Snowpits and SD measurements taken before and after (± 2 days) each TSX acquisition were included in the analysis as no precipitation occurred and air temperature was stable during the field campaign.***

**Reviewer comment 5:** Observation period covers many days - does each acquisition include data from many days? 24 scenes – how do the snow pits correspond with these? This needs a bit more clarification. It is clear the historical data used for the time series is from Orbit 24, but how many of the 104 scenes are in the 2019 period like the others?

Thank you for the comment. TerraSAR-X has a repeat orbit of 11 days. For each orbit, we have a continuous time series during the winter season with acquisitions every 11 days / with data gap from March 6th 2018 to April 17th 2019. A total of five scenes were used for the regression statistics. An average of 10 snowpits for each day were conducted for each day, and two snow depth transects were acquired on May 1st, 2019 (please refer to Transect#2 in Fig.4) and May 4th 2019 (Transect #1). Following modification in the manuscript are added (lines 147-149):

***Additionally, two SD transects were conducted across the catchment to analyze the SD distribution in the study site. Both transects were established from the east side to the west side of the Ice Creek catchment. These transect were acquired on May 1st 2019 (Transect #2) and May 4th 2019 (Transect #1).***

**Reviewer comment 6:** 3.2.1 – two snow pit characterization sampling strategies? Unclear what the second is. the revisited pits vs the ones elsewhere in the catchments?

We apologize for the confusion. The sampling strategy is explained in line 140-147:

"The snowpit locations in the centre of the Ice Creek catchment as well as location at the outlet of the catchment were revisited *after* each TerraSAR-X (TSX, see 3.3.) acquisition so that soil characteristics remain unchanged between snow sampling and satellite measurements. Snow depths were measured using a GPS snow depth probe around the snowpits, ensuring the representativeness of the snowpit location.

[...] Snowpits and SD measurements were then distributed spatially elsewhere in the catchment to refine the characterization of snow within the catchment."

**Reviewer comment 7:** Lines 303-305: Why the variability in the CPD annual mean? Is that an acceptable range of annual variability to group into an overall 2014-19 mean? Or is there any comparison that can be made. Presumably it's related to different weather conditions year-to-year? But is there anything you can conclusively say about what is driving that? perhaps I missed that, but it would be interesting to know if the range in annual means can be explained by e.g. snow depth. Perhaps not as 14-15 and 17-18 years highlighted as the min and max are also the years missing from the climate data.

Well observed - Sentences on line 303-305 were meant to report seasonal values over each year. The wording seems to confuse the readers so here is the modification:

"For the 2015-2019 period, the mean CPD value during the snow season was -8.59°. ***The mean of each winter is ranging between 13.41° (2014-2015) and -6.42° (2017-2018).*** During the snow-free condition, the average CPD over the same period increased to -0.87° (2015-2019). ***Maximum and minimum values during snow-free conditions ranged between -0.44° (2015) and -1.32° (2015-2016). "***

**Reviewer comment 8:** Objective 2 – I don't think you fully met this objective. You do show the temperature and precipitation data with the CPD, but is anything else examined? (see comment below about table 1). Nothing is gone into in depth in this section directly relating the meteorological data to the CPD other than the cyclical pattern between snow and non snow times. You should consider making more use of the data to explore the meteorological links or revise the wording of objective 2.

Thank you for the hint. After consideration, we suggest to re-wording the objectives and add a third one (modification at lines 64-67):

(1) investigate SD and DHF variability between different vegetation classes in the Ice Creek catchment (Qikiqtaruk-Herschel Island, Yukon, Canada) using in situ measurement collected over the course of a field campaign in 2019; *(2) evaluate linkages between snow characteristics and CPD distribution over 2019 dataset and (3) Determine CPD seasonality considering meteorological data over 2015-2019 period.*

**Reviewer comment 9:** A few clarifications and very minor typos that caught my eye as I was reading:

Something that would be useful is a sentence early on clarifying that the winter year you report in some places refers to the previous fall and next years spring (e.g. 2018 means 2017-2018)

Thank you for the hint. Yes, the winter refers to the period from mid-September to mid-May, as written at section 4.2.1., lines 298-299:

"The period with presence of snow was set between mid-September and mid-May based on prior observations (Burn and Zhang, 2009; Stettner et al., 2018)."

To avoid any confusion and keep consistency, line 305-306 was changed: The decrease generally started in January, when the average air temperature is at its coldest (-20 °C) *except during the snow season of 2016-2017,* where a warming occurred, increasing the average temperature of 5 °C for that year.

**Reviewer comment 10:** Figure 1: define delta rho

Sorry for the confusion. Δρ in Figure 1 defines the phase shift. We clarified it in the text and in the figure.

**Reviewer comment 11:** Figure 2: text line 148 says east to west transects, figure shows west to east transect?

We are sorry, that was a typo. We revised the sentence as follows (line 148): *"Both transects were established from the west side to the east side of the Ice Creek catchment."*

**Reviewer comment 12:** Figure 4: caption, depth shouldn't be capitalized, also you refer to it as depth hoar fraction elsewhere and ratio here

We revised Figure 4 caption and we changed "depth hoar ratio" to "depth hoar fraction" - thank you for the advice.

**Reviewer comment 13:** Figure 5: "is shows" is shown

Thank you. We corrected it.

**Reviewer comment 14:** "Windows pixels size is 1x1 pixel (5x5m)" – explain?

We use the CPD mean value within a window of 5x5 m. We have chosen a smaller window size to better reflect the heterogeneity of the snow surface, which could alter within a few meters. The high heterogeneity of the snowpack and the need for its monitoring at the landscape scale motivate our approach at the finer resolution possible, as discussed in the introduction:

"Current snow modules used in Earth System Models are based on coarse spatial resolution of tens of kilometres (Bokhorst et al., 2016). Coarse spatial resolution hampers our efforts to understand the dynamics driving snowpacks at the landscape scale. Indeed, snow is characterized by a high spatial and temporal heterogeneity (e.g.: Rutter et al., 2014; Thompson et al., 2016; Wilcox et al., 2019). Traditional approaches using *in situ* measurement can provide very detailed spatial information on snow properties, but cannot be deployed over large areas. There is therefore a strong need to bridge these two scales and provide means to monitor the temporal and spatial variability of the snowpack over larger areas."

The linear regression analysis showed good results at this scale so we decided to keep this resolution.

**Reviewer comment 15:** Any established relationships between the temp/precip at the Herschel Station and somewhere on the north slope Alaska, or Tuktoyaktuk, That you can use to fill the gaps? Or is the weather too unique to the island? Might be worth a sentence clarifying that the nearby stations are not related, or too far to be used. Alternatively, is there a need to
exclude the entire winter from the chart when some months are available? (end of 2017 is missing, but early 2018 is available – or is the data flawed?).

Thank you for the suggestion. We filled the air temperature gap of the meteorological station on Herschel Island using data from Komakuk Beach as performed by Burn and Zhang (2009). Following equation was used to apply on Komakuk dataset:

$T_h = 0.97 \times T_k + 0.75$

Where $T_h$ are the monthly mean air temperature at Herschel island and $T_k$ at Komakuk Beach. Predicted values showed good correlation (R2:0.93, p-value = < 0.001) with RMSE of 3.32 °C (Fig.1).

[Figure]

*Fig. 1:* *result from linear regression between air temperature measured on Herschel Island and predicted value using Burn and Zhang 2009 equation.*

Figure 2 show a visual comparison between the air temperature predicted and measured along the time series. Therefore, the gaps will be filled using this equation. Figure 5 was changed consequently (see Fig. 3) and Fig 1 and 2 from this document are added in the appendix.

[Figure]

*Fig. 2:* *Comparison during 1994-2022 air temperature measured on Herschel Island and predicted value using Burn and Zhang 2009 equation.*

Unfortunately, no precipitation datasets were available at Komakuk Beach station between 2014-2019. The final figure is shown below, where the red dotted line shows the gap filled data. Modification in the legend is also showed:

[Figure]

**Figure 5.** (a) Average CPD and (b) Average CCOH by vegetation class with interval of confidence (95%) for orbit 24 (31°, descending). Pixels values were extracted from GPS dataset (see Fig. 2c), where $N_{Coltsfoot}$ = 33, $N_{Dryas}$ = 140, $N_{Lupine}$ = 118, $N_{Shrub}$ = 29. Winter period (mid-September to mid-May) is shown in shaded area. Windows pixels size is 1x1 pixel (5x5m). (c) Meteorological data from Qikiqtaruk Herschel Island station (dataset from Environment Canada (2021)). The meteorological station is not equipped with a telemetry system and since the island is inaccessible during the winter, the lack of data during the winters 2014-2015 and 2017-2018 was caused by a malfunction of the station***. Air temperature during theses periods were gap filled using Komakuk Beach meteorological station and shown by the red dotted line.. Please refer to appendix A for further details on the method. .***

**Reviewer comment 16:** Figure 6: snow depth on the x axis should be 2 words based on how its written elsewhere in the manuscript

Corrected.

**Reviewer comment 17:** Table 1: are you using the ECCC wind and humidity data for anything? Why list just those variables from the tower (and does the datalogger really matter)? If they are used, considering adding a column to the table for what they are used for.

You are right. As this information is not relevant for the understanding of the work, we deleted it.

**Reviewer comment 18:** Line 33: snow depth trend[s] (s missing)

We corrected it.

**Reviewer comment 19:** Line 44: spatial (says special)

Thank you! We corrected it accordingly.

**Reviewer comment 20:** Line 49: (possible elsewhere?) Spaceborne is typically one word

Corrected.

**Reviewer comment 21:**Line 73: just terminology here, consider explaining kinetic metamorphic regime for the more general reader? Presumably, most readers will be snow scientists or related fields and understand this, but a few sentences to clarify would be helpful for those who are not.

Thank you for the suggestion. We revised the sentence as followed (line 74):

***"Kinetic growth refers to formation of depth hoar within the snowpacks induced by a strong thermal gradient".***

**Reviewer comment 22:** Line 83: "…vertical direction after their setting up in the snowpack". While I understand what you mean here, the wording reads a little odd to me. Is 'setting up' in the snowpack common terminology? Consider a more formal explanation here?

You are right. We modified the sentence as follows (line 82-83):

"As such, over the time, snow crystals become elongated to a vertical direction ***after and during the constructive snow metamorphosis in the snowpack"***

**Reviewer comment 23:**Line 87-88: "such as used in this study or below as dry snow can be" – I think this sentence needs a comma, I believe you are referring to shorter wavelengths, but I did have to reread this one a few times.

Thank you for the correction. We added the comma (line 87-88):

"Given the dry nature of the arctic snowpack, the main source of backscattering should occur at the snow-ground interface for frequencies in X-band ($\lambda$ = 3.1 cm), such as used in this study, or below as dry snow can be considered as a homogeneous, "non-scattering" and non-absorbing volume (Leinss et al., 2014). "

**Reviewer comment 24:** Line 93: random? vs. randomly

We corrected it to "random phase shift".

**Reviewer comment 25:**Line 131: you probably need the full citation for ArcticDem here as well as other online data sets mention later

ArticDEM is cited and referred as Porter et al., 2018. The reference below respects the journal's guideline:

" Porter, C., Morin, P., Howat, I., Noh, M.-J., Bates, B., Peterman, K., Keesey, S., Schlenk, M., Gardiner, J., Tomko, K., Willis, M., Kelleher, C., Cloutier, M., Husby, E., Foga, S., Nakamura, H., Platson, M., Wethington Michael, J., Williamson, C., Bauer, G., Enos, J., Arnold, G., Kramer, W., Becker, P., Doshi, A., D'Souza, C., Cummens, P., Laurier, F. and Bojesen, M.: ArcticDEM [data set], doi:10.7910/DVN/OHHUKH, 2018."

**Reviewer comment 26:** Line 158: snow depth [and] mean density

Corrected

**Reviewer comment 27:** Line 256: SD decreased significantly? It was tested? Or this is just a word choice here? If I'm missing somewhere her with the stats test, no problem,

but otherwise pick something that does not imply a stats test since you have several stats tests also being talked about. Substantially, or greatly, or …

Thank you for the correction. We modified the sentience as followed:

*"Further west, SD values decreased substantially on the slope with values between 20 and 50 cm."*

**Reviewer comment 28:** Line 258: "Along transect #2 (Fig. 4), and snow cover" remove , and

Corrected. Thank you for your contribution.

---

## Author Comment (AC3)

**Potential of X-band polarimetric SAR co-polar phase difference for Arctic snow depth estimation**
**tc-2021-314**

**Revisions**

Joëlle Voglimacci-Stephanopoli, Anna Wendleder, Hugues Lantuit, Alexandre Langlois, Samuel Stettner, Andreas Schmitt, Jean-Pierre Dedieu, Achim Roth, and Alain Royer

Reviewer's comments
Answers to reviewer
***Modification to text***

Dear Reviewer,

Thank you very much for the useful comments and suggestions regarding our work. This is highly appreciated. For a better readability of our response, the corrections in the manuscript are shown in orange.

**Reviewer comments 1:** Line 58: "The main challenge related to the use of SAR is the lack of a reliable method to relate satellite data to physical measurements in snow impacted environments." Would the authors be able to expand on why they see other SAR based methods unreliable? Some studies to provide examples:

Lievens, H. et al. (2019) 'Snow depth variability in the Northern Hemisphere mountains observed from space', Nature communications, 10(1), p. 4629.
Eppler, J. and Rabus, B. T. (2021) 'The Effects of Dry Snow on the SAR Impulse Response and Feasibility for Single Channel Snow Water Equivalent Estimation', IEEE transactions on geoscience and remote sensing: a publication of the IEEE Geoscience and Remote Sensing Society, pp. 1–23.

Alternatively, the sentence can be reworded to clarify the intended meaning.

The sentence will be removed.

Lievens et al. (2019) found a method to derive globally the snow depth in alpine areas at 1km$^2$ resolution. Our research focused more on snow characterization in arctic snowpack with consideration to the landscape scale (hence, at higher resolution).

Eppler and Rabus (2021) show the feasibility of the L-band VV polarization in L-band to estimate the SWE. Eppler and Rabus's paper was published last summer and seems to have gone under our radar while we finished this manuscript. This is indeed an interesting method which need to be studied further with spaceborne as their experiments come from airborne data. They actually addressed this point in this paper under review:

Eppler, J., Rabus, B. T., and Morse, P.: Snow Water Equivalent Change Mapping from Slope Correlated InSAR Phase Variations, The Cryosphere Discuss. [preprint], https://doi.org/10.5194/tc-2021-359, in review, 2021.

**Reviewer comments 2:** Line 185, Figure 3: Please consider adding a map of the TWI to give the readers that are unfamiliar with the area an understanding of how it varies

over the study area.

Thank you. We suggest to add a TWI map from our study site in the appendix so the reader will have access to the figure if needed. Here the map of the TWI and its legend added in appendix B.

[Figure]

**Figure B2: Topographic wetness index (TWI) map compared to vegetation units located on Qikiqtaruk-Herschel island.**

**Reviewer comments 3:** Line 210: Were the CCOH calculated directly from the 5m resolution Kennaugh elements?
If so, isn't it a biased estimator at that resolution? For reference Leinss et al. 2014, was using an averaging window of about 75m.

The reviewer is right, Silvan Leinss smoothed the SAR signal using filter sizes of 75 m. We discussed that point with him and his recommendation was to try different filter sizes and to analyze which filter size fits the best. We have chosen a smaller window size to better reflect the heterogeneity of the snow surface which could alter within a few meters. Regarding the smoothing, the original spatial resolution of 2.5 m was resampled to 5 m before geocoding. Additionally, the SAR images were smoothed after geocoding with an enhancement approach called multi-scale multi-looking which adapts the local number of looks to the image content.

**Reviewer comments 4:** No-Line: Given that the CPD = 2*pi*SD/wavelength*birefringent_refractive_index, and that the birefringent_refreactive_index is about 1deg/cm of snow per Leinss et al., 2014, wouldn't there be cases where phase unwrapping may be needed for deep snow (~ >180cm)? If so, how was this handled? If not, why was it not needed?

Phase unwrapping is applied in case of continuous gradients which, for example, is the case for the generation of elevation models using InSAR. Though, the seasonal evolution of snow is not continuous as the CPD depends not only on the snow height but also on snow metamorphism. An indication when phase unwrapping should be applied is the existence of fringes in the diagram which is not the case (neither in our analyses nor in those of Silvan Leinss).

**Reviewer comment 5:** No-Line: Have the authors considered the phase noise of TanDEM-X sensor ( +/-3.5 deg Leinss et al., 2014) in their uncertainty analysis? If not, why was it not needed?

Good point! The phase noise refers to absolute measurements. Our analyses are based on relative phase measurements. Therefore, there is no need to consider the phase noise.

**Reviewer comment 6:** Minor:
L93: "randomly phase shifts"
Consider "randomly shifting phase"
After verifications, we corrected it to "random phase shift".

L122: "reach over a 110 cm"
Consider "reach over 110cm"
L412: " hummocky area "
consider "hummocky areas"

Thank you. All changes were made.

---

## Author Response (AR2)

Response to Reviewers' comments to manuscript tc-2021-314 – 2[nd] revision

**"Potential of X-band polarimetric SAR co-polar phase difference for Arctic snow depth estimation"**
* * *
Dear Editor and Reviewer,

Thank you for your careful review.

The point-by-point response to the second iteration of revision is attached to this response.

Reviewer's comments
Answers to reviewer
*Modification in the text*

Line 23: "DHF" is not defined in the abstract

We included  "Depth hoar fraction (DHF) in the abstract.

line 26: remove "n."
Line 68-69: change "2019 and (2)" to "2019, (2)". Lower case "d" after (3)
Line 100: The two parts of this sentence are disconnected. Should there be an "and" or "or" after the comma?
Line 155: "transect(s)", plural

Thank you. All the corrections were made.

Line 161: I don't think "depth hoar" needs to be capitalized. Are "melt-freeze crust" and "ice layer" considered two separate layer classifications, or the same?

That was a typo. Following changes were made:

"From the above observations, each layer was classified according to their density and snow grain type across *6 classes* following Fierz et al. (2009): 1) depth hoar, 2) windslab, 3) surface hoar, 4) fresh snow, 5) melt-freeze crust and *6) ice layer"*

Line 164: remove comma after "classified"

Thank you. We corrected it.

Line 227: Depth Hoar Fraction (DHF) is never really described. Suggest adding the equation or a description here

Following sentence was added in the paragraph:

"We focused on the SD variability between vegetation classes. We also evaluated depth hoar fraction (DHF) given that King et al. (2018) found that X-band backscattering is highly sensitive to depth hoar grains. *DHF is the depth hoar ratio within the total depth of a snowpit.* "

Section starting at line 235, "Topographic Wetness Index as a proxy": You don't say what TWI is a proxy for. It's a little confusing since this section talks about snow surface properties and also water content at the soil surface, as well as vegetation classes.

We agree with the reviewer. Subtitle was change to *Topographic wetness index* and following modification were made in line 241-242:

The Topographic Wetness Index (TWI) was chosen to analyze the variance between vegetation groups *as a potential indicator of variability*.

Line 259: "371 pixels were used", instead of "was"

Thank you. We corrected it.

Figure 5 caption: "Windows pixels size is 1x1 pixel (5x5m)." This wording is still confusing – it's not clear what you mean by "windows". Could this be something like, "The window over which vegetation class information was extracted is the same size as a TSX pixel (5x5m)"?

Thank you. The sentence was deleted and changed to:

*"The window over which vegetation class information was extracted is the same size as a TSX pixel (5x5m)."*

Line 437: missing a word between "allows" and "to"

Sentence in lines 441-443 was changed to:

*"With a* high incidence angle ($> 30°$) *and* a high TWI ($> 7.0$)*, significant correlation* between SD and CPD *can be found* with *a* R-squared *of* 0.72. "